# ZHX2 promotes HIF1α oncogenic signaling in triple-negative breast cancer

Wentong Fang[1,2†], Chengheng Liao[3*†], Rachel Shi[3], Jeremy M Simon[2,4], Travis S Ptacek[2,5], Giada Zurlo[3], Youqiong Ye[6], Leng Han[7], Cheng Fan[2], Lei Bao[3], Christopher Llynard Ortiz[8,9,10], Hong-Rui Lin[8], Ujjawal Manocha[2], Weibo Luo[3], Yan Peng[3,11], William Y Kim[2], Lee-Wei Yang[8,9,12], Qing Zhang[3*]

[1]Department of Pharmacy, The First Affiliated Hospital of Nanjing Medical University, Nanjing, China; [2]Lineberger Comprehensive Cancer Center, University of North Carolina School of Medicine, Chapel hill, United States; [3]Department of Pathology, University of Texas Southwestern Medical Center, Dallas, United States; [4]Department of Genetics, Neuroscience Center; University of North Carolina School of Medicine, Chapel Hill, United States; [5]UNC Neuroscience Center, Carolina Institute for Developmental Disabilities, University of North Carolina, Chapel Hill, United States; [6]Shanghai Institute of Immunology, Faculty of Basic Medicine, Shanghai Jiao Tong University School of Medicine, Shanghai, China; [7]Department of Biochemistry and Molecular Biology, The University of Texas Health Science Center at Houston McGovern Medical School, Houston, United States; [8]Institute of Bioinformatics and Structural Biology, National Tsing Hua University, Hsinchu, Taiwan; [9]Chemical Biology and Molecular Biophysics Program, Taiwan International Graduate Program, Institute of Chemistry, Academia Sinica, Taiwan; [10]Department of Chemistry, National Tsing-Hua University, Hsinchu, Taiwan; [11]Harold C. Simmons Comprehensive Cancer Center, University of Texas Southwestern Medical Center, Dallas, United States; [12]Physics Division, National Center for Theoretical Sciences, Hsinchu, Taiwan

*For correspondence:
chengheng.liao@utsouthwestern.edu (CL);
qing.zhang@utsouthwestern.edu (QZ)

†These authors contributed equally to this work

**Abstract** Triple-negative breast cancer (TNBC) is an aggressive and highly lethal disease, which warrants the critical need to identify new therapeutic targets. We show that Zinc Fingers and Homeoboxes 2 (*ZHX2*) is amplified or overexpressed in TNBC cell lines and patients. Functionally, depletion of ZHX2 inhibited TNBC cell growth and invasion in vitro, orthotopic tumor growth, and spontaneous lung metastasis in vivo. Mechanistically, ZHX2 bound with hypoxia-inducible factor (HIF) family members and positively regulated HIF1α activity in TNBC. Integrated ChIP-seq and gene expression profiling demonstrated that ZHX2 co-occupied with HIF1α on transcriptionally active promoters marked by H3K4me3 and H3K27ac, thereby promoting gene expression. Among the identified ZHX2 and HIF1α coregulated genes, overexpression of *AP2B1*, *COX20*, *KDM3A*, or *PTGES3L* could partially rescue TNBC cell growth defect by *ZHX2* depletion, suggested that these downstream targets contribute to the oncogenic role of ZHX2 in an accumulative fashion. Furthermore, multiple residues (R491, R581, and R674) on ZHX2 are important in regulating its phenotype, which correspond with their roles on controlling ZHX2 transcriptional activity in TNBC cells. These studies establish that ZHX2 activates oncogenic HIF1α signaling, therefore serving as a potential therapeutic target for TNBC.

## Editor's evaluation

Zinc-fingers and homeoboxes (ZHX) has been highlighted as one critical hypoxia-related factors regulator contributing to triple negative breast cancer in this work, which has enriched the upstream

regulatory network of HIF-1α signaling. Moreover, identification of key residuals determining biological function of ZHX2 provides novel approaches for treating TNBC via targeting hypoxia pathway.

## Introduction

Triple-negative breast cancer (TNBC) accounts for 15–20% of all breast cancer (*Anders and Carey, 2009*). TNBC is associated with a more aggressive clinical history, a higher likelihood of distant metastasis, shorter survival, and a higher mortality rate compared to other subtypes of breast cancer (*Anders and Carey, 2009*). In addition, recent studies illustrate high rates of brain metastasis in TNBC that is associated with poor survival (*Heitz et al., 2009*; *Lin et al., 2008*; *Niwińska et al., 2010*). Since TNBCs do not express estrogen receptor (ER), progesterone receptor (PR), or human epidermal growth factor receptor 2 (HER2), treatment options have historically been limited to chemotherapy (*Masui et al., 2013*), which has significant toxicity and a suboptimal impact on the 5-year relapse rate. Therefore, it is critical to identify novel therapeutic targets in TNBC.

The Zinc Fingers and Homeoboxes (*ZHX*) family includes *ZHX*1, 2, and 3. *ZHX2* is located on 8q24.13 and contains two zinc finger domains and five homeodomains (HDs) (*Kawata et al., 2003*). In addition, between amino acids 408 and 488, it contains a proline-rich region (PRR). ZHX2 can form homodimers or can heterodimerize with the other two family members ZHX1 or ZHX3 (*Kawata et al., 2003*). Originally, ZHX2 was found to be a key transcriptional repressor for the alpha-fetoprotein regulator 1 (*Afr1*) (*Perincheri et al., 2005*), which is an important oncogene in liver cancer. From this perspective, ZHX2 was identified and reported to function as a transcriptional repressor (*Kawata et al., 2003*), where fusion of ZHX2 with a GAL4-DNA-binding domain repressed transcription of a GAL4-dependent luciferase reporter. Additionally, ZHX2 was reported to have tumor suppressor activity in hepatocellular carcinoma (HCC), by repressing cyclin A, E, and multidrug resistance 1 (MDR1) expression (*Ma et al., 2015*; *Yue et al., 2012*). ZHX2 was also indicated to be a tumor suppressor in Hodgkin lymphoma or myeloma although there is no direct experimental evidence supporting this hypothesis (*Armellini et al., 2008*; *Nagel et al., 2012*; *Nagel et al., 2011*).

However, accumulating evidence suggests that ZHX2 may contribute to cancer pathology in other contexts. Tissue microarray and clinicopathological analysis show that ZHX2 protein expression in metastatic HCC is twice as high as in the primary lesions, indicating that ZHX2 expression is associated with metastasis in HCC (*Hu et al., 2007*). In addition, our recent findings through genome-wide screening identify that ZHX2 is a substrate of von Hippel Lindau (gene name *VHL*, protein name pVHL) protein, accumulates in kidney cancer, and promotes oncogenic signaling by at least partially activating nuclear factor κB (NF-κB) signaling in clear cell renal cell carcinoma (ccRCC) (*Zhang et al., 2018*). These pieces of evidence suggest that ZHX2 acts as a tumor suppressor or oncogene in a context-dependent manner. It is also important to point out that *ZHX2* is located on 8q24, a genomic region that is frequently amplified in various cancers including breast cancer (*Guan et al., 2007*). More importantly, the role of ZHX2 in other cancers, such as in TNBC, remains largely unknown.

Tumor hypoxia is a characteristic of most solid tumors. Hypoxic cells are known to confer radio- or chemotherapeutic resistance, and therefore are hypothesized to undergo positive selection during cancer development (*Brown and Wilson, 2004*; *Gray et al., 1953*). The key proteins mediating oxygen sensing in these cells involve two classes of proteins: (1) upstream oxygen sensors, namely the prolyl hydroxylases EglN1–3, responsible for the hydroxylation of various substrates, such as hypoxia-inducible factor (HIF), FOXO3a, ADSL, SFMBT1, and TBK1 (*Hu et al., 2020*; *Liu et al., 2020*; *Semenza, 2012*; *Zheng et al., 2014*; *Zurlo et al., 2019*); (2) the downstream VHL E3 ligase complex. For example, EglN family members (EglN1, primarily in vivo) hydroxylate HIF1α on proline 402 and 564 positions, which lead to pVHL binding and HIF1α ubiquitination and degradation (*Appelhoff et al., 2004*; *Ivan et al., 2001*; *Jaakkola et al., 2001*). HIF1α has been well established to be an important oncogene in multiple cancers, including breast cancer (*Briggs et al., 2016*; *Semenza, 2010*). Tumor hypoxia or pVHL loss will lead to the accumulation of HIF1α. As a result of the accumulation and translocation of HIFα factors into the nucleus, HIF1α dimerizes with a constitutively expressed HIF1β subunit (also called ARNT) and transactivates genes that have hypoxia response elements (HREs, sequence: NCGTG) in promoters or enhancer regions. HIF1-transactivated genes include those involved in angiogenesis (e.g., *VEGF*), glycolysis and glucose transport (e.g., *SLC2A1*),

and erythropoiesis (e.g., *EPO*) (*Semenza, 2012*). Besides tumor hypoxia or pVHL loss, other potential regulators of HIF1α may exist, which remains to be investigated.

In our current study, we investigated the role of *ZHX2* as a new oncogene in TNBC, where it activates HIF1α signaling. In addition, we also provide some evidence for critical residues on ZHX2 that binds with DNA, which contributes to TNBC tumorigenicity.

## Results

### ZHX2 is amplified in TNBC and is potentially regulated by pVHL

*ZHX2* is located on 8q24, where *MYC* (gene product: c-Myc) resides. Analysis of copy number across different cancer types from The Cancer Genome Atlas (TCGA) showed that *ZHX2* is amplified in various cancers, including ovarian cancer (~40%) and breast cancer (~15%) (*Figure 1A*). Importantly, *ZHX2* and *MYC* share coamplification in most cancer types observed (*Figure 1B*). Interestingly, *ZHX2* is not amplified in ccRCC (referred as KIRC in *Figure 1A,B*), where it is regulated mainly post-transcriptionally by pVHL loss in this cancer (*Zhang et al., 2018*). Detailed analyses of several breast cancer patient datasets also revealed that TNBC had the highest *ZHX2* amplification rate in all breast cancer subtypes (*Figure 1C*, *Supplementary file 1a*). Next, we performed correlation studies to examine the copy number gain of *ZHX2* and its expression in TCGA (*Cancer Genome Atlas Network, 2012*) and METABRIC datasets (*Curtis et al., 2012 Figure 1D*). We found a significant correlation between copy number and gene expression (*Figure 1D*), suggesting that *ZHX2* amplification may be at least partially responsible for its overexpression in breast cancer. We further explored the effect of ZHX2 expression on breast cancer patient survival. *ZHX2* overexpression (Affymetrix probe 236169_at) correlates with worse survival in TNBC but not in other breast cancer subtypes (*Figure 1—figure supplement 1A, B*). We obtained a panel of breast cancer cell lines as well as two immortalized normal breast epithelial cell lines, HMLE and MCF-10A. Interestingly, all breast cancer cell lines displayed relatively higher ZHX2 protein levels compared to HMLE or MCF-10A (*Figure 1E*). We detected ZHX2 protein level in 10 pairs of TNBC patient tumors and their respective normal control (*Supplementary file 1b*). Consistently, ZHX2 was upregulated in the majority of tumors compared to normal (8 out of 10) (*Figure 1F*). To strengthen the clinical significance of ZHX2, we also performed immunohistochemistry (IHC) staining in 20 pairs of normal/tumor TNBC patient parafilm tissue slides and found an overall increased ZHX2 staining in the TNBC tumors, especially in the nuclear staining by quantification (*Figure 1G, H*), suggesting overall and the nuclear located ZHX2 is upregulated in tumors from TNBC patients.

Our previous research established an oncogenic role of ZHX2 in ccRCC as a pVHL substrate (*Zhang et al., 2018*). However, the role of ZHX2 in other cancers remains largely unclear. In addition, it is unclear whether ZHX2 also acts as a pVHL target in breast cancer. Since TNBC had the highest ZHX2 amplification rate in all breast cancer subtypes (*Figure 1C*, *Supplementary file 1a*), we decided to focus on TNBC for this current study. To this purpose, we first examined the relationship between the expression of ZHX2 and pVHL in several TNBC cell lines (MDA-MB-231, MDA-MB-436, MDA-MB-468, HCC3153, and HCC70) as well as two normal breast epithelial cell lines, HMLE and MCF-10A. Interestingly, all TNBC cell lines displayed relatively lower pVHL protein levels corresponding with higher ZHX2 protein level when compared to normal breast cells (*Figure 1—figure supplement 1C*). Our co-immunoprecipitation (Co-IP) experiments showed that ZHX2 interacts with pVHL in two representative TNBC cell lines (*Figure 1I*). Next, we aimed to examine whether pVHL can promote the degradation of ZHX2, therefore decreasing ZHX2 protein levels in these cells. First, we overexpressed HA-VHL in two different TNBC cell lines (MDA-MB-231 and 468) and found that pVHL overexpression leads to decreased ZHX2 protein levels (*Figure 1J*). Conversely, we also deleted *VHL* expression by three independent sgRNAs (#1, 2, and 8) in these two cell lines and found that pVHL depletion led to upregulation of ZHX2 protein (*Figure 1K*, *Figure 1—figure supplement 1D*). Our previous research showed that ZHX2 regulation by pVHL potentially depends on ZHX2 prolyl hydroxylation (*Zhang et al., 2018*). We treated these two cell lines with hypoxia, the pan-prolyl hydroxylase inhibitor DMOG, or the proteasomal inhibitor MG132 and found that ZHX2 was upregulated by these treatments (*Figure 1L*, *Figure 1—figure supplement 1E*), further strengthening the conclusion that ZHX2 is regulated by pVHL for protein stability through potential prolyl hydroxylation in breast cancer. We further analyzed ZHX2 and pVHL protein levels in 10 pairs of TNBC tumors and paired normal

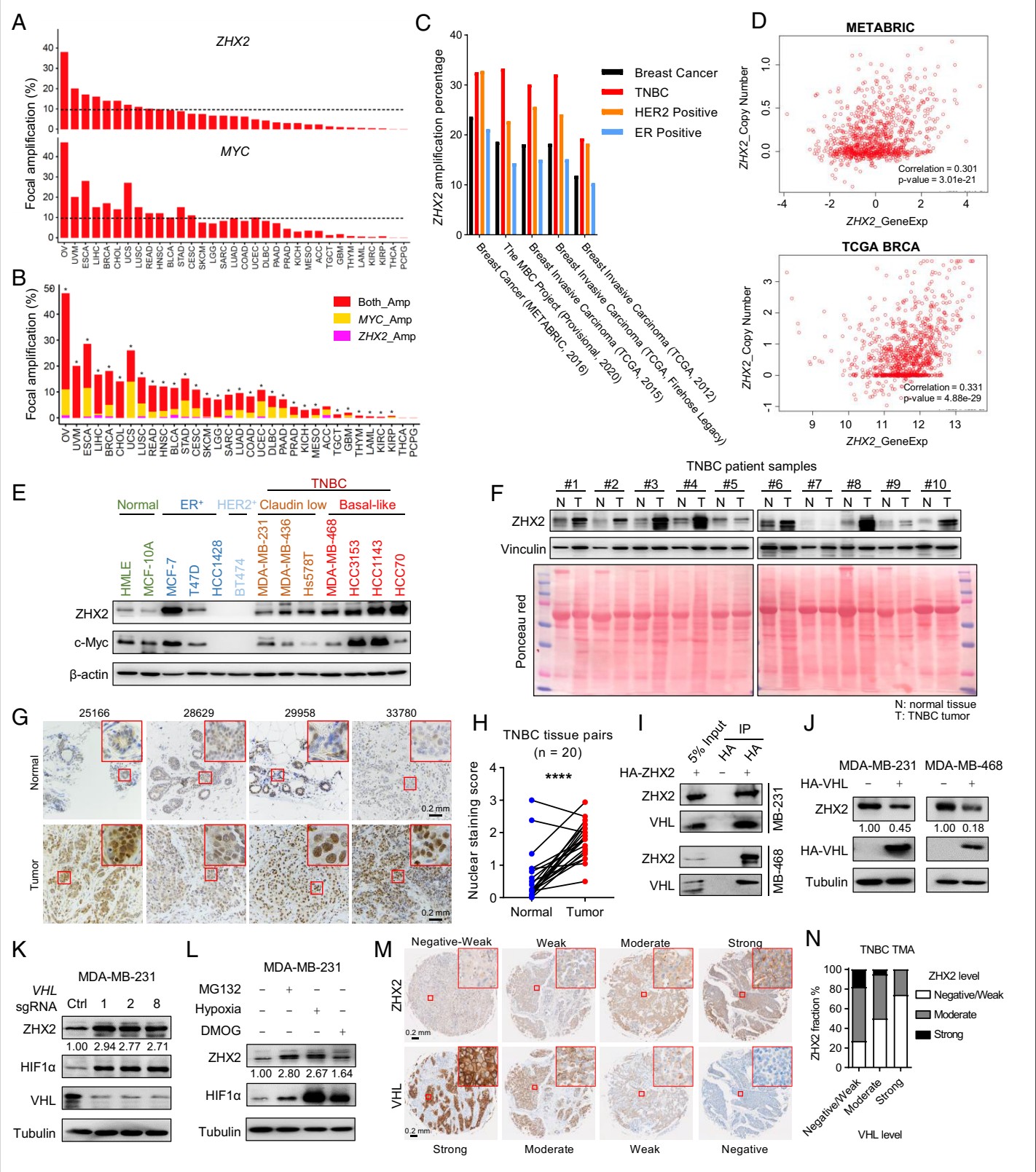

**Figure 1.** *ZHX2* is amplified in triple-negative breast cancer (TNBC) and is potentially regulated by pVHL. (**A**) The percentage of tumor samples with *ZHX2* (top) or *MYC* (bottom) focal amplification across cancer types. (**B**) The percentage of samples with both *ZHX2* and *MYC* (red), *ZHX2* specific (magenta), and *MYC* specific (orange). Asterisk indicates statistical significance for overlap of *ZHX2* and *MYC* focal amplification (Fisher's exact test, p < 0.05). (**C**) The percentage of *ZHX2* amplification of different breast cancer subtypes in several breast cancer datasets. (**D**) The relation of *ZHX2* copy

*Figure 1 continued on next page*

*Figure 1 continued*

number gain and its expression in The Cancer Genome Atlas (TCGA) datasets and METABRIC datasets. (**E**) Immunoblots of lysates from different normal breast cell and breast cancer cell lines. (**F**) Immunoblots of lysates from paired TNBC patient-derived normal (N) and tumor (T) breast tissues. (**G, H**) Representative immunohistochemistry (IHC) staining images (**G**) and quantification (**H**) of ZHX2 protein level in paired TNBC patient-derived normal and tumor breast tissues. (**I**) Immunoprecipitations (IP) and immunoblot of MDA-MB-231 and 468 cell lysates with HA beads. (**J**) Immunoblot of cell lysates from MDA-MB-231 and 468 cells stably expressed control vector or HA-*VHL*. (**K**) Immunoblot of cell lysates from MDA-MB-231 infected with lentivirus encoding either *VHL* sgRNAs (1, 2, or 8) or control sgRNA (Ctrl). (**L**) Immunoblots of lysates from MDA-MB-231 cells treated with hypoxia or indicated inhibitors for 8 hr. (**M, N**) Representative IHC staining images (**M**) and quantification (**N**) of human TNBC specimens with four staining grades showing the expression correlation between ZHX2 and VHL protein. Error bars represent mean ± standard error of the mean (SEM), unpaired *t*-test. ****p < 0.0001.

The online version of this article includes the following figure supplement(s) for figure 1:

**Source data 1.** Uncropped western blot images for *Figure 1*.

**Figure supplement 1.** ZHX2 overexpression leads worse patient survival and is potentially regulated by pVHL in breast cancer.

**Figure supplement 1—source data 1.** Uncropped western blot images for *Figure 1—figure supplement 1*.

tissue (*Supplementary file 1b*). In accordance with the cell line data, we found that ZHX2 was upregulated in most of tumor tissues compared to normal, coinciding with decreased pVHL protein levels in respective tumor tissues (*Figure 1—figure supplement 1F*). To further examine the clinical relevance of pVHL with ZHX2, we obtained two sets of commercial TNBC tissue microarray (TMA, BR1301) from US Biomax, which contains 130 human TNBC specimens. We performed IHC staining to the TMA with ZHX2 and pVHL antibody and scored the IHC staining intensities (*Detre et al., 1995*) of each specimen for these two proteins, respectively. We found an overall negative correlation between pVHL and ZHX2 in these TNBC samples which were divided into different staining grades based on the staining intensity (*Figure 1M, N* and *Figure 1—figure supplement 1G*). In conclusion, our data suggest that ZHX2 is regulated by pVHL and play an important role in TNBC.

## ZHX2 is essential for TNBC cell proliferation and invasion

Next, we examined the potential role of ZHX2 in TNBC cell proliferation and invasion. First, we obtained two previously validated *ZHX2* shRNAs (sh43 and sh45) (*Zhang et al., 2018*) and these shRNAs led to efficient ZHX2 protein (*Figure 2A*) and mRNA (*Figure 2B*) downregulation in both MDA-MB-231 and MDA-MB-468 cells. Next, we found that ZHX2 depletion led to decreased cell proliferation, 2D colony formation as well as 3D soft agar growth in both cell lines (*Figure 2C–F*). One important contributor for the poor prognosis in TNBC is the aggressively invasive nature of this subtype. Therefore, we also used the Boyden chamber assay to examine the effect of ZHX2 on cell invasion in TNBC cells. Consistent with the results above, ZHX2 depletion led to decreased cell invasion in these TNBC cell lines (*Figure 2G, H*). We also showed that *ZHX2* shRNA-induced phenotypes in TNBC cells could be completely rescued by shRNA-resistant *ZHX2* in two TNBC cell lines (*Figure 2I–N* and *Figure 1—figure supplement 1A–G*), suggesting that these phenotypes were due to on-target depletion of ZHX2 by shRNAs.

## ZHX2 is important for TNBC cell proliferation in vivo

Next, to examine the ability of ZHX2 to maintain TNBC tumor growth in vivo, we first constructed firefly luciferase stably expressing MDA-MB-231 cells (MDA-MB-231-luc) followed by infecting these cells with doxycycline (Dox)-inducible control or *ZHX2* shRNAs (Teton sh43, sh45). Upon Dox addition, we achieved efficient depletion of ZHX2 protein (*Figure 3A*). ZHX2 depletion upon Dox addition led to consistent decreased TNBC cell proliferation, 2D growth, and 3D anchorage-independent growth of MDA-MB-231 in vitro (*Figure 3—figure supplement 1A–D*). Next, we injected the *ZHX2* Teton shRNAs or control shRNA expressing MDA-MB-231-luc cells orthotopically into the fourth mammary fat pads of NoD SCID Gamma (NSG)-deficient mice. After palpable tumors were formed after implantation, we fed these mice Dox chow. We found that ZHX2 depletion significantly decreased tumor growth overtime (*Figure 3B*), as well as reduced tumor burden retrieved from tumor-bearing mice compared to control mice upon necropsy (*Figure 3C, D*). We also performed a western blot for samples extracted from the MDA-MB-231-derived xenograft tumors and found decreased ZHX2 protein levels in *ZHX2* Teton shRNA-infected groups (*Figure 3E*), arguing that antitumor effect in these groups may result from efficient ZHX2 knockdown. Given the high in vivo lung metastatic feature of MDA-MB-231

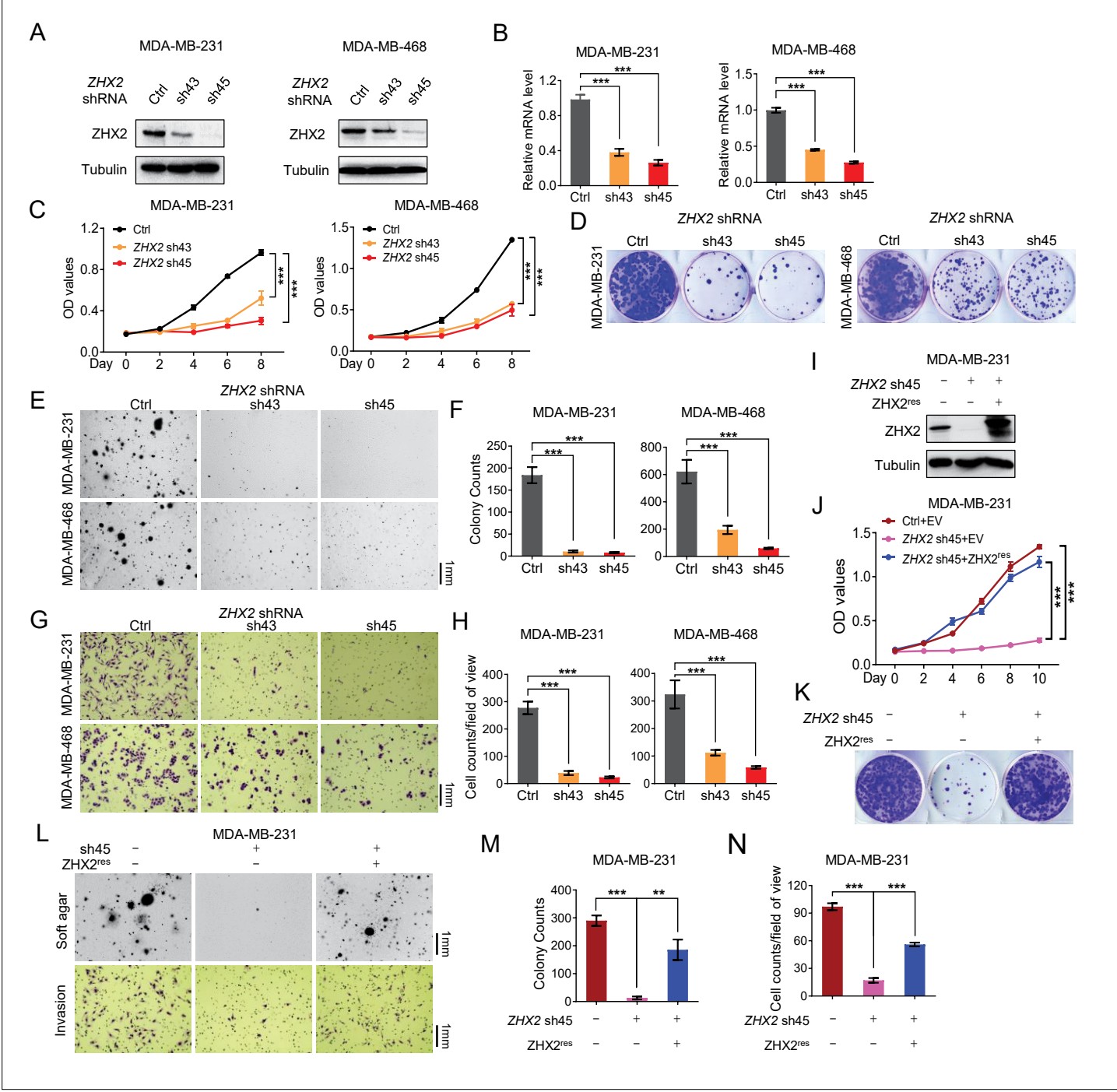

**Figure 2.** ZHX2 is essential for triple-negative breast cancer (TNBC) cell proliferation and invasion. (**A–H**) Immunoblot of cell lysates (**A**), Real-time quantitative PCR (qRT-PCR) (**B**), cell proliferation assays (**C**), 2D colony formation assays (**D**), representative images of 3D soft agar growth (**E**) and quantification of oft agar colonies (**F**), invasion assay (**G**), and quantification of invasion assay (**H**) of MDA-MB-231/468 cells infected with lentivirus encoding two individual *ZHX2* shRNAs (43 and 45) or control shRNA (Ctrl). (**I–N**) Immunoblot of cell lysates (**I**), cell proliferation (**J**), 2D colony formation assay (**K**), representative images of 3D soft agar growth (upper), and invasion assays (down) (**L**) as well as their quantifications (**M, N**) of MDA-MB-231 cells infected with lentivirus encoding sh45-resistant *ZHX2* (*ZHX2*^res^) or control vector (EV), then followed by *ZHX2* sh45 or Ctrl lentivirus infection. Error bars represent mean ± standard error of the mean (SEM), unpaired *t*-test. **p < 0.01; ***p < 0.001.

The online version of this article includes the following figure supplement(s) for figure 2:

**Source data 1.** Uncropped western blot images for *Figure 2*.

**Figure supplement 1.** The phenotype of *ZHX2* shRNA on cell proliferation and invasion is due to its on-target effect.

**Figure supplement 1—source data 1.** Uncropped western blot images for *Figure 2—figure supplement 1*.

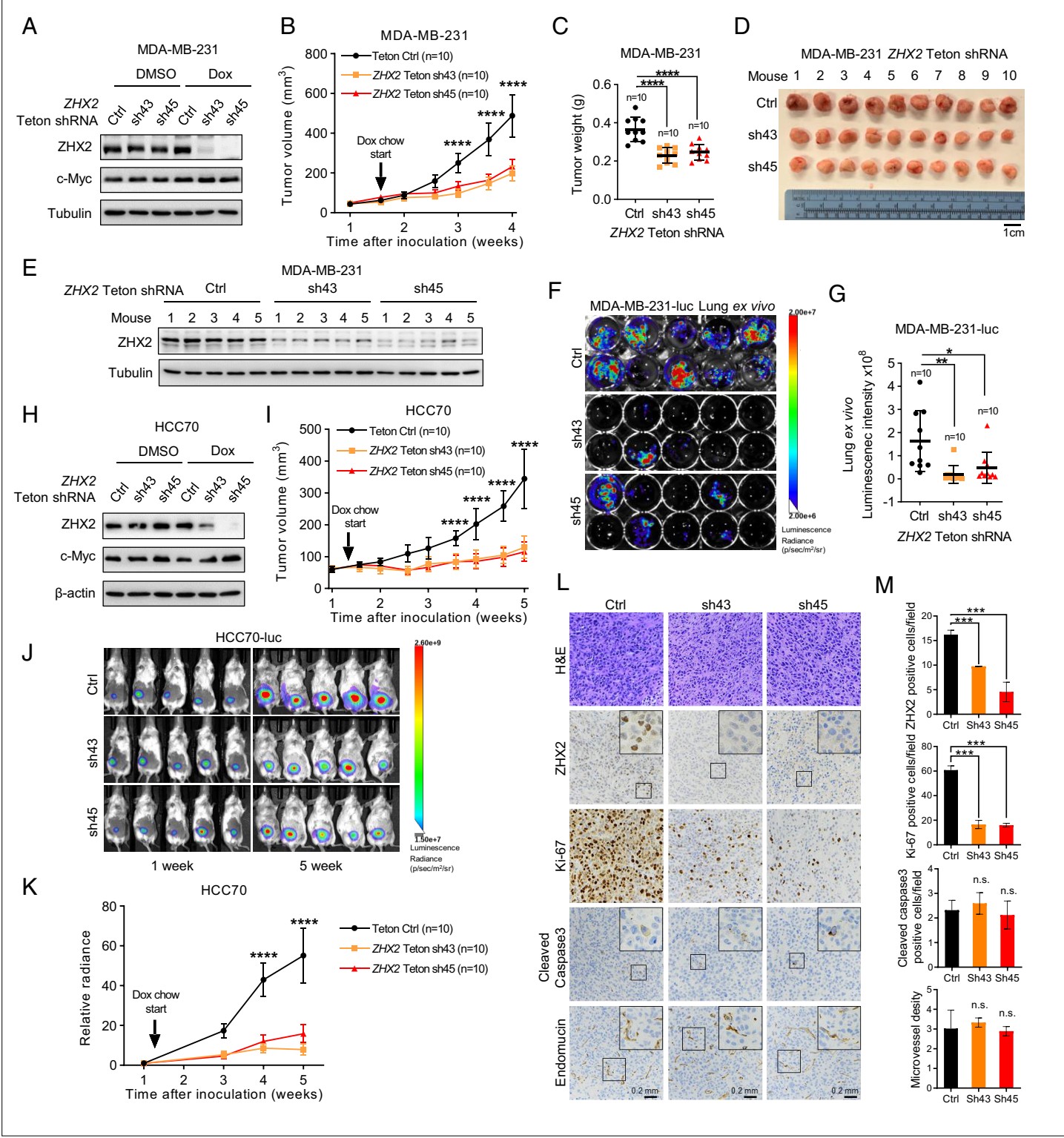

**Figure 3.** ZHX2 is important for triple-negative breast cancer (TNBC) cell proliferation and tumorigenesis in vivo. (**A, B**) Immunoblot of cell lysates (**A**) and tumor growth (**B**) of MDA-MB-231-luc cells expressing doxycycline (Dox)-inducible control or *ZHX2* shRNAs (Teton Ctrl, sh43, 45). Cells were orthotopically injected at the mammary fat pad of NoD SCID Gamma (NSG) mice. Treatment of Dox food started as indicated. (**C, D**) Tumor weight (**C**) and image (**D**) from MDA-MB-231 xenografts mice after dissection. (**E**) Immunoblot of lysates from the MDA-MB-231 xenograft tumors. (**F, G**) Lung ex vivo bioluminescence imaging (**F**) and quantification of lung ex vivo imaging (**G**) of the MDA-MB-231-luc xenograft mice. (**H, I**) Immunoblot of cell lysates (**H**) and tumor growth (**I**) of HCC70-luc cells expressing Dox-inducible control or *ZHX2* shRNAs (Teton Ctrl, sh43, 45). Cells were

*Figure 3 continued on next page*

*Figure 3 continued*

orthotopically injected at the mammary fat pad of NSG mice. Treatment of Dox food started as indicated. (**J, K**) Representative bioluminescence images at the indicated weeks (**J**) and quantification of the bioluminescence imaging (**K**) of the HCC70-luc xenograft tumors. (**L, M**) Representative immunohistochemistry (IHC) staining images (**L**) and quantification (**M**) of hematoxylin–eosin (H&E), ZHX2, Ki-67, cleaved caspase 3, and Endomucin in tumors (*n* = 4) from the MDA-MB-231 xenograft tumors. Error bars represent mean ± standard error of the mean (SEM), unpaired *t*-test. *p < 0.05; **p < 0.01; ***p < 0.001; ****p < 0.0001.

The online version of this article includes the following figure supplement(s) for figure 3:

**Source data 1.** Uncropped western blot images for *Figure 3*.

**Figure supplement 1.** ZHX2 is important for maintaining triple-negative breast cancer (TNBC) tumorigenesis in vivo.

**Figure supplement 1—source data 1.** Uncropped western blot images for *Figure 3—figure supplement 1*.

(*Jin et al., 2020*), we measured spontaneous lung metastasis ex vivo upon necropsy and found that ZHX2 depletion led to significantly decreased lung metastasis in MDA-MB-231 cells (*Figure 3F, G*). Additionally, we also orthotopically injected *ZHX2* Teton sh45-infected MDA-MB-231 cells, followed by control or Dox chow. Consistent with results above, Dox chow led to significantly decreased TNBC tumor growth over time and decreased ZHX2 protein levels in tumors (*Figure 3—figure supplement 1E–H*). To confirm these results in another TNBC cell line, we also generated the inducible control or *ZHX2* shRNAs in HCC70-luc cells and performed the in vivo studies. We found consistent results in terms of ZHX2 knockdown (*Figure 3H*), in vitro growth (*Figure 3—figure supplement 1I–K*), as well as in vivo tumor growth (*Figure 3I*, *Figure 3—figure supplement 1L, M*). In addition to monitor tumor growth by caliper, we also performed live bioluminescence imaging over time and provide representative image of initial and end of the experiment for the HCC70-luc xenograft mice (*Figure 3J*), we found that the relative bioluminescence signal intensity is consistent with the tumor volume growth (*Figure 3K*). Lastly, we took four tumors from each group from the MDA-MB-231 xenograft tumors followed by hematoxylin–eosin (H&E) staining as well as IHC staining with ZHX2, Ki-67 (proliferative marker), Endomucin (angiogenic marker), and cleaved-caspase 3 (apoptotic maker) antibodies. IHC assays showed that ZHX2 depletion mainly affected proliferation for the tumor cells reflected by marked decrease of Ki-67 in ZHX2 depleted tumor, rather than angiogenesis or apoptosis (*Figure 3L and M*). Although ZHX2 and c-Myc displayed similar amplification and expression manner in breast cancer (*Figure 1A, B, E*). Our data showed that knockdown ZHX2 did not affect protein level of c-Myc (*Figure 3A, H*), suggesting the oncogenic function of ZHX2 is independent of c-Myc in TNBC cells. Taken together, our data strongly indicate that ZHX2 is important for TNBC cell proliferation and tumorigenesis in vivo.

## ZHX2 regulates HIF signaling in TNBC

Next, we aimed to determine the molecular mechanism by which ZHX2 contributes to TNBC. We previous find that ZHX2 promoted NF-κB activation by directly binding with p65 and promoting its nuclear translocation in ccRCC (*Zhang et al., 2018*). We first examined the interaction of ZHX2 and p65 in breast cancer by endogenous IP with ZHX2 or p65 antibody in two TNBC cells (MDA-MB-231 and 468) (*Figure 4—figure supplement 1A, B*), we did not find consistent and robust binding between ZHX2 and p65 as we observed in ccRCC. In addition, knockdown of ZHX2 also did not change the gross p65 protein level (*Figure 4—figure supplement 1C*), or the subcellular protein level of p65 either in cytosol or nucleus (*Figure 4—figure supplement 1D*, E). Altogether, these data suggest ZHX2 may not affect NF-κB signaling in TNBC.

In order to determine the molecular mechanism by which ZHX2 contributes to TNBC, we performed RNA-seq analyses in MDA-MB-231 cells with two independent *ZHX2* shRNAs (sh43 and sh45) and found that two individual shRNAs concordantly altered gene expression patterns (*Figure 4A, B*). Pathway enrichment as well as gene set enrichment analyses (GSEAs) revealed that genes differentially expressed following ZHX2 depletion were enriched for members of the hypoxia pathway (*Figure 4C* and *Figure 4—figure supplement 2A–C*). Consistent with the lack of regulation of NF-κB components by ZHX2 by IP and western blots, we did not find the NF-κB pathway being enriched through our RNA-seq analysis pathway (*Figure 4C* and *Figure 4—figure supplement 2A*). Overall, there were 7690 genes differentially expressed following *ZHX2* silencing by shRNA, and 1849 (24%) of these genes overlapped with genes differentially expressed in a previously published *HIF1A* and

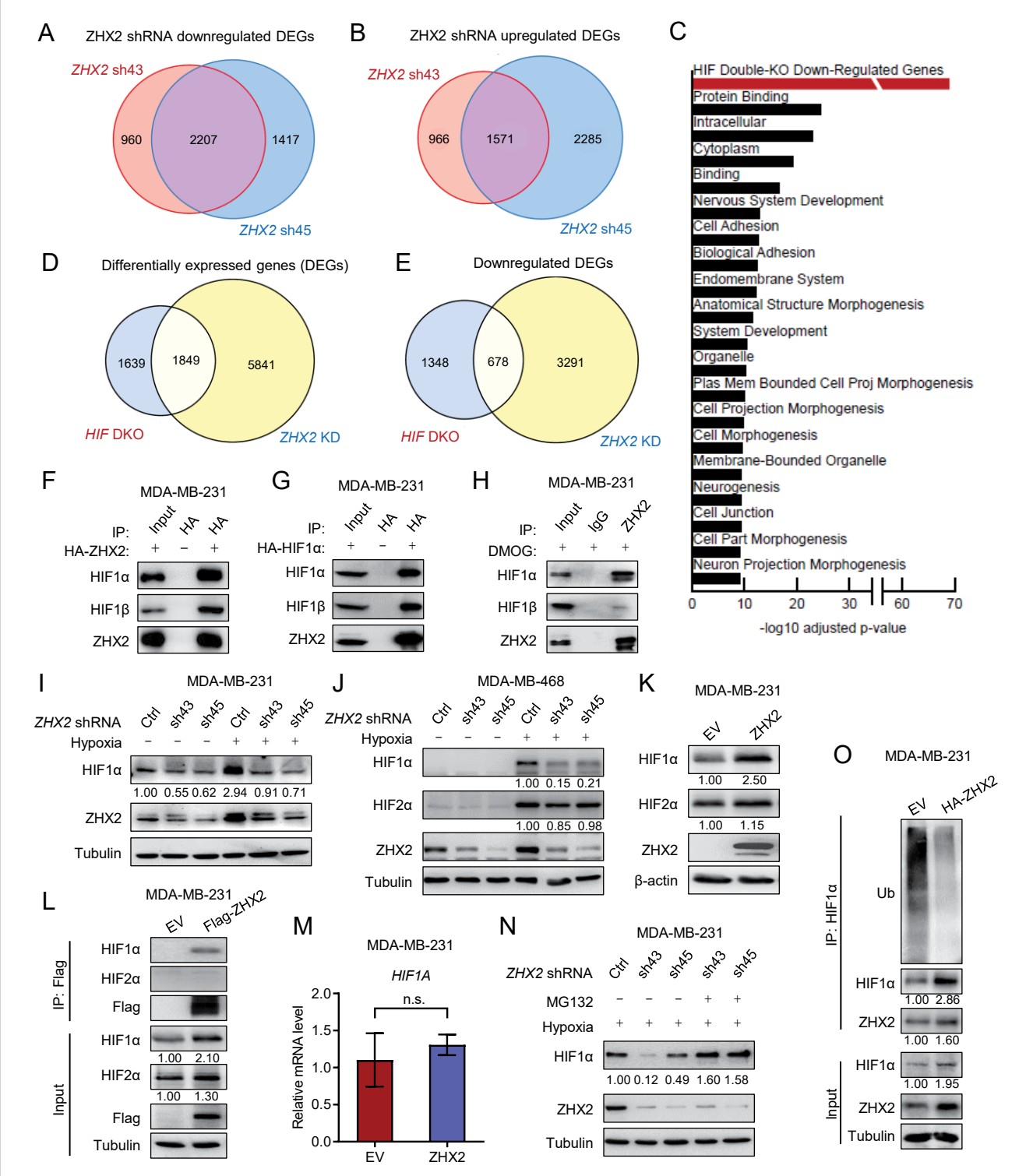

**Figure 4.** ZHX2 regulates hypoxia-inducible factor (HIF) signaling in triple-negative breast cancer (TNBC). (**A, B**) Venn diagrams showing the overlap in downregulated (**A**) or upregulated (**B**) differentially expressed genes (DEGs) between *ZHX2* shRNA 43 and 45. (**C**) Pathway analysis of the significantly decreased pathways in *ZHX2* depleted MDA-MB-231 cells. (**D, E**) Venn diagrams showing overlap in DEGs (**D**) and downregulated DEGs (**E**) between *ZHX2* depletion and *HIF* double knockout (DKO) (GSE108833). (**F, G**) Immunoprecipitations (IP) and immunoblots with indicated antibodies of MDA-MB-231 cells overexpress either HA-ZHX2 (**F**) or HA-HIF1α (**G**). IP was performed with HA beads. (**H**) IPs of MDA-MB-231 cells treated with DMOG for 8 hr. (**I, J**) Immunoblots of cell lysates from MDA-MB-231 cells (**I**) and MDA-MB-468 cells (**J**) infected with lentivirus encoding *ZHX2* shRNAs or Ctrl, followed by treating with normoxia or hypoxia (1% O$_2$). (**K–M**) Immunoblots (**K**) and IPs (**L**) of cell lysates, qRT-PCR of mRNA (**M**) from MDA-MB-231 cells

*Figure 4 continued on next page*

*Figure 4 continued*

infected with lentivirus encoding control vector (EV) or ZHX2. (**N**) Immunoblots of cell lysates from MDA-MB-231 cells infected with lentivirus encoding *ZHX2* shRNAs or Ctrl treated with MG132 overnight under hypoxia (1% O$_2$). (**O**) IPs of cell lysates from MDA-MB-231 cells infected with lentivirus encoding control vector (EV) or HA-ZHX2. Error bars represent mean ± standard error of the mean (SEM), unpaired *t*-test. n.s., not significant.

The online version of this article includes the following figure supplement(s) for figure 4:

**Source data 1.** Uncropped western blot images for *Figure 4*.

**Figure supplement 1.** ZHX2 does not affect p65 translocation in triple-negative breast cancer (TNBC).

**Figure supplement 1—source data 1.** Uncropped western blot images for *Figure 4—figure supplement 1*.

**Figure supplement 2.** ZHX2 regulates hypoxia-inducible factor (HIF)1 signaling in triple-negative breast cancer (TNBC).

**Figure supplement 2—source data 1.** Uncropped western blot images for *Figure 4—figure supplement 2*.

*HIF2A* double knockout (*HIF* DKO) model (*Figure 4D*; *Chen et al., 2018*). Since ZHX2 was similarly reported as an oncogene in ccRCC and preferentially upregulated the transcription of downstream genes (*Zhang et al., 2018*), we focused on the 3969 ZHX2 positively regulated genes (downregulated following ZHX2 silencing by shRNA) as these genes may be more relevant in breast cancer. Of these, 678 genes (17%) overlapped with downregulated genes in the *HIF* DKO RNA-seq (*Figure 4E*), representing a significant association (adj. p = 1.07 × 10$^{-69}$). These data strengthen the potential functional link between HIF and ZHX2. The genes positively regulated by ZHX2 were also enriched for other potentially relevant biological pathways such as cell adhesion and cell morphogenesis (*Figure 4C*). Next, we examined whether ZHX2 depletion affected some of the canonical HIF target genes. Based on our ZHX2 RNA-seq and GSEA results (*Figure 4—figure supplement 2C*), we chose a few representative HIF target genes and examined their expression levels by qRT-PCR in MDA-MB-231 cells infected with control or *ZHX2* shRNA under either normoxia or hypoxia. As expected, hypoxia treatment increased the expression of HIF target genes, and this effect was ameliorated by ZHX2 depletion (*Figure 4—figure supplement 2D*), suggesting that ZHX2 affects HIF activity and regulates HIF target gene expression in TNBC. Consistently, HIF reporter assay found that ZHX2 depletion led to decreased HIF reporter activity while ZHX2 re-expression could rescue this phenomenon either under normoxia or hypoxia (*Figure 4—figure supplement 2E*). In addition, ZHX2 activated HIF reporter activity in a dose-dependent manner (*Figure 4—figure supplement 2F*).

Next, to gain further insight on how ZHX2 affect the HIF signaling. Consider a well characterization on HIF1 function in TNBC tumorigenesis by previous studies (*Bos et al., 2001*; *Briggs et al., 2016*). We first performed Co-IP experiments and showed that ZHX2 bound with HIF1α and HIF1β exogenously as well as endogenously (*Figure 4F–H*, *Figure 4—figure supplement 2G*). In addition, ZHX2 knockdown led to decreased HIF1α protein levels in two TNBC cell lines under hypoxia condition (*Figure 4I, J*). Conversely, ZHX2 overexpression led to increased HIF1α protein levels under normoxia (*Figure 4K*). However, these ZHX2 loss-of-function or gain-of-function manipulations did not grossly change HIF2α protein level (*Figure 4J, K*). Co-IP experiments showed that ZHX2 could not bind with HIF2α (*Figure 4L*). These results suggest that ZHX2 regulates the hypoxia signaling mainly through HIF1α. Interestingly, qRT-PCR showed that ZHX2 overexpression did not increase *HIF1A* mRNA level (*Figure 4M*), suggesting a mechanism of post-transcriptional regulation on HIF1α. We then treated the ZHX2 knockdown cells with the proteasome inhibitor MG132 under hypoxia, which could fully rescue the HIF1α protein level in the knockdown cells (*Figure 4N*). Our ubiquitination assay also showed that overexpression of ZHX2 could protect HIF1α from ubiquitination and subsequent degradation (*Figure 4O*). Altogether, these data suggested ZHX2 controls HIF1α protein stability by preventing proteasome-mediated degradation. On the other hand, western blot analysis showed that HIF1α knockdown did not affect ZHX2 protein levels (*Figure 4—figure supplement 2H*). It is important to note that the detailed mechanism on how ZHX2 binds with HIF1 and transactivates HIF1 signaling remains unclear, which awaits future investigation.

To identify the direct downstream target genes of ZHX2 and HIF that may be important in TNBC, we performed chromatin IP followed by high-throughput sequencing (ChIP-seq) to assess the genomic binding pattern of ZHX2 in TNBC. We identified 957 binding sites across the genome, of which 94% of them overlap with H3K27ac and 96% of them overlap with H3K4me3 (*Figure 5A*), indicating these overlapping peaks bound preferentially to active promoters for gene expression (*Shlyueva et al., 2014*). We again focused on the ZHX2 positively regulated genes (downregulated following *ZHX2*

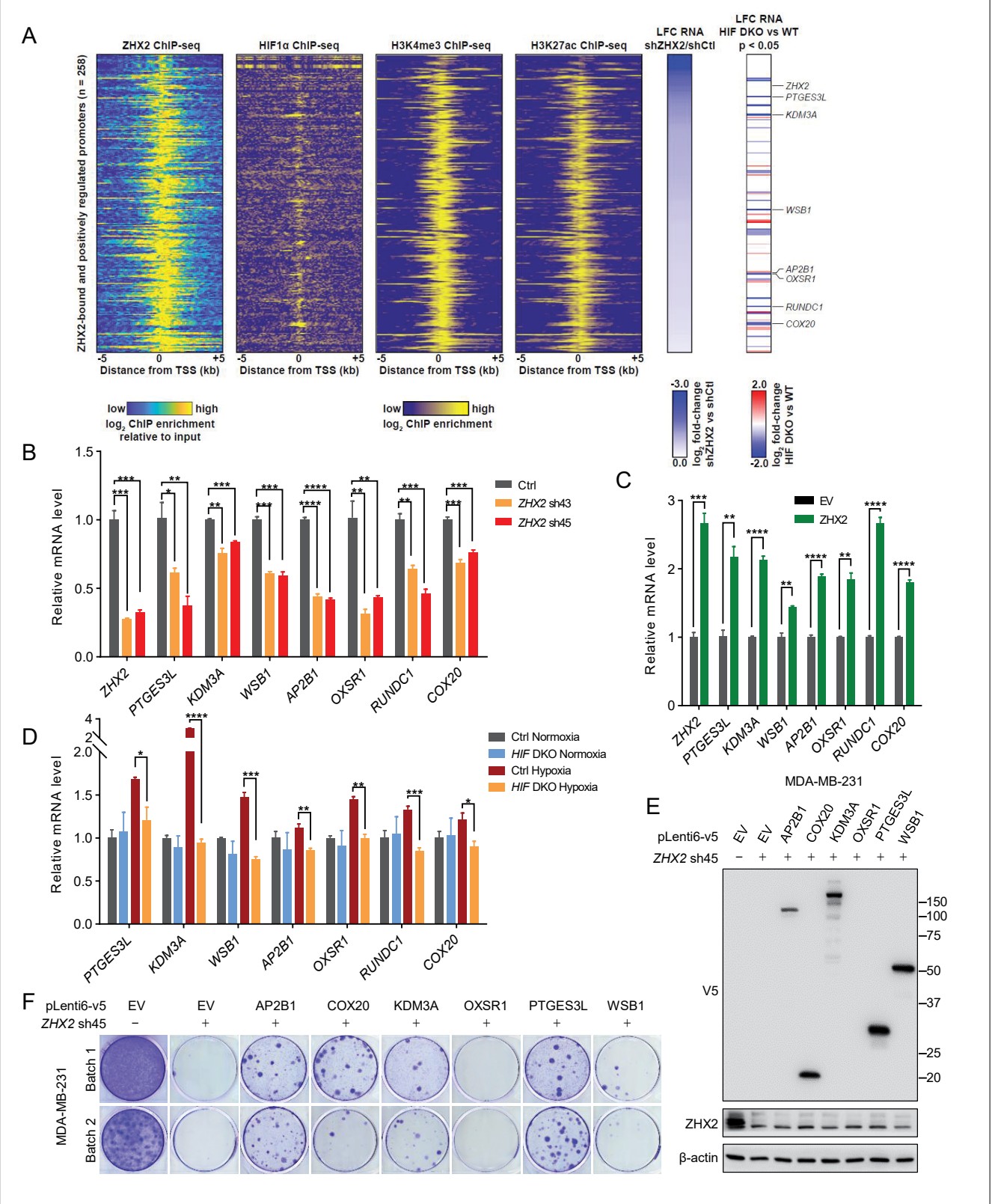

**Figure 5.** Representative ZHX2 and HIF downstream targets and analysis of their chromatin-binding motifs. (**A**) Integrated analyses of ChIP-seqs (including ZHX2, HIF1α, H3K4me3, and H3K27ac), signals expressed as relative to input control when available. Log2 fold change (LFC) for *ZHX2* knockdown RNA-seq and *HIF* double knockout (*HIF* DKO) RNA-seq; critical target genes were marked on the right. (**B–D**) qRT-PCR quantification of ZHX2 target genes from MDA-MB-231 cells infected with indicated lentivirus encoding *ZHX2* shRNAs (43 and 45) (**B**), control vector (EV) or ZHX2 (**C**), or

*Figure 5 continued on next page*

*Figure 5 continued*

*HIF* DKO under normoxia (21% O₂) or hypoxia (1% O₂) (**D**). (**E**) Representative immunoblots of cell lysates from MDA-MB-231 cells infected with lentivirus encoding EV or *AP2B1*, *COX20*, *KDM3A*, *OXSR1*, *PTGES3L*, and *WSB1*, and followed by depletion of *ZHX2* by sh45. (**F**) 2D colony formation growth of stable selected cells from (**E**). Batach1 and 2 indicate two biological experiments. Error bars represent mean ± standard error of the mean (SEM), unpaired *t*-test. *p < 0.05; **p < 0.01; ***p < 0.001; ****p < 0.0001.

The online version of this article includes the following figure supplement(s) for figure 5:

**Source data 1.** Uncropped western blot images for *Figure 5*.

silencing by shRNA) that exhibited ZHX2 binding in the promoter (transcription start site ±5 kb). These promoters (n = 258) demonstrated robust enrichment for HIF1α (*Chen et al., 2018*), as well as H3K4me3 and H3K27ac (*Rhie et al., 2014*; *Figure 5A*). We then filtered these genes further, focusing on only those bound and positively regulated by HIF1α and identified seven interesting candidate genes for follow-up functional validation (*PTGES3L*, *KDM3A*, *WSB1*, *AP2B1*, *OXSR1*, *RUNDC1*, and *COX20*). We performed qRT-PCR analysis and found that *ZHX2* depletion by shRNAs indeed led to decreased mRNA expression for all of these target genes (*Figure 5B*). Conversely, we also overexpressed ZHX2 in TNBC cells and found that ZHX2 overexpression led to increased expression of these target genes (*Figure 5C*), arguing that ZHX2 promotes HIF signaling by at least directly activating these targets in TNBC. To further strengthen whether these are HIF downstream target genes, we also obtained *HIF* DKO cells and found that HIF depletion led to decreased expression of these target genes under hypoxic conditions (*Figure 5D*). Taken together, our data suggest that ZHX2 binds with HIF and affects HIF activity in TNBC.

To determine which gene is essential for maintaining the oncogenic function of ZHX2, we performed rescue experiments by overexpressing some of these downstream target genes. We obtained six lentivirus-based expression constructs (*PTGES3L*, *KDM3A*, *WSB1*, *AP2B1*, *OXSR1*, and *COX20*) from DNASU (https://dnasu.org/DNASU/Home.do). Upon generating stable blasticidin selected cell lines for each gene in MDA-MB-231, we subsequently knocked down *ZHX2* by sh45 in each stable cell lines. We eventually found that five genes expressed with expected size except OXSR1 in these selected stable cell lines (*Figure 5E*). We then performed 2D colony formation assays and found that at least four genes (*AP2B1*, *COX20*, *KDM3A*, and *PTGES3L*) could partially rescue the growth by ZHX2 depletion (*Figure 5F*). As expected, the cell line failed to express OXSR1 could not rescue the phenotype. The partially rescue phenotype suggested that these downstream targets contribute to the oncogenic role of ZHX2 in an accumulative fashion.

## Potential important sites on ZHX2 that may affect its function in TNBC

Previous research showed that ZHX2 contributed to ccRCC tumorigenesis by at least partially activating NF-κB signaling (*Zhang et al., 2018*). In that setting, ZHX2 may act as transcriptional activator by overlapping primarily with H3K4me3 and H3K27Ac epigenetic marks. However, it remains unclear whether there may be critical residues on ZHX2 that may mediate its binding to DNA and exert its transcriptional activity. To this end, we conducted both data-driven analyses and structural simulations to predict the essential DNA-binding residues. Detailed methods are described in *Supplementary file 1*. Briefly, many DNA-bound structures of human HD2, 3 and 4 (PDB ID: 3NAU, 2DMP, and 3NAR) were homology modeled by SWISS-MODEL (*Waterhouse et al., 2018*) using as many X-ray/NMR-solved homologous DNA-bound HD proteins as the structural templates. 12, 20, and 15 DNA-bound complexes were modeled for HD2, HD3, and HD4, respectively (*Figure 6—figure supplement 1*). We then counted the number of DNA–protein contacts at the atomic level for each residue in the DNA-contacting helices in HDs, normalized by the number of

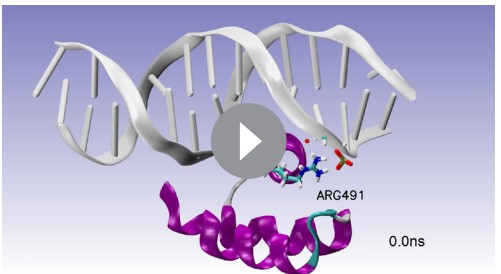

**Video 1.** The 350-ns molecular dynamics (MD) simulation trajectory of 1NK2-based HB2–dsDNA complex. The HB2–dsDNA complex is in secondary structure cartoon representation. The top C-terminal helix residue that most contacts the DNA, which is ARG491, and the DNA atoms within 4 Å distance from it are in licorice representation.

https://elifesciences.org/articles/70412/figures#video1

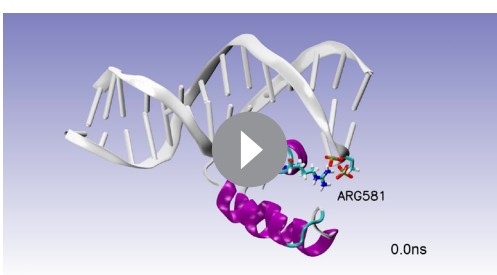

**Video 2.** The 350-ns molecular dynamics (MD) simulation trajectory of 1MNM-based HB3–dsDNA complex. The HB3–dsDNA complex is in secondary structure cartoon representation (violet color: HB3's helix; gray color: dsDNA). The top C-terminal helix residue that most contacts the DNA, which is ARG581, and the DNA atoms within 4 Å distance from it are in licorice representation.

https://elifesciences.org/articles/70412/figures#video2

DNA-complexed structures used for each HD. The top-ranked DNA-contacting residues in HDs are listed together with their evolutionary conservation in *Supplementary file 1c*. We found that Lys485/Arg491 in HD2, Arg581 in HD3, and Arg674 in HD4 are the most contacted residues in DNA binding, where Arg674 receives the highest contact among all the HD proteins. Our molecular dynamics (MD) simulations further revealed that Arg491 in HD2, Arg581 in HD3, and Arg674 in HD4 indeed have the highest affinity with DNA, in terms of molecular mechanics Poisson–Boltzmann surface area (MM/PBSA)-derived contact potential energies (see *Supplementary file 1d,e Videos 1–3*), among other residues in the same proteins. Among the top 4 DNA-contacting residues in each of the HD proteins, we consider those with relatively high sequence conservation being important for ZHX2 binding to DNA, which may affect the phenotype of TNBC. These residues are Asp489 (D489), Arg491 (R491), Glu579 (E579), Arg581 (R581), Lys582 (K582), Arg674 (R674), Glu678 (E678), and Arg680 (R680) (*Supplementary file 1c*).

Given this, we generated a series of TNBC breast cancer cell lines where we depleted endogenous ZHX2 expression and restored with exogenous shRNA-resistant *ZHX2* WT or mutant versions (D489A, R491A, E579A, R581A, K582, R674A, E678A, or R680A). First, upon generation of stable cell lines, we performed western blot analyses and confirmed that these cell lines all expressed similar amounts of ZHX2, relatively comparable to endogenous ZHX2 levels in MDA-MB-231 cells (*Figure 6A*). Next, we performed 2D cell proliferation MTS assays. Our cell proliferation data showed that consistent with our previous results, ZHX2 depletion led to decreased TNBC cell proliferation, and this phenotype was completely rescued by WT ZHX2. On the other hand, some of mutants (including R581A and R674A) failed to rescue the cell growth defect in *ZHX2* shRNA-infected MDA-MB-231 cells (*Figure 6B*).

Motivated by our cell proliferation assay results, we further examined cell proliferation phenotypes using long-term 2D colony formation, 3D anchorage-independent growth, and cell invasion assays. ZHX2 R491A, R581A, and R674A mutants displayed a defect in cell growth in TNBC cell lines (*Figure 6C–F*). In summary, our results showed that there may be multiple residues (including R491, R581, and R674) that may be important in regulating the phenotype of ZHX2 in TNBC. To examine these ZHX2 functional residues will affect HIF1α activity, our Co-IP analysis showed that mutated ZHX2 (R581A and R674A) did not affect ZHX2 interaction with HIF1α (*Figure 6G*). In addition, these ZHX2 mutants led to increased HIF1α protein levels to the similar extent as wild type ZHX2 (*Figure 6H*), while qRT-PCR showed that the ZHX2 mutations indeed led to decreased selective mRNA expression of HIF1α targeted genes (*Figure 6I*). Taken together, our data suggest that these residues are essential for mediating the transactivation of ZHX2 on HIF1α target genes.

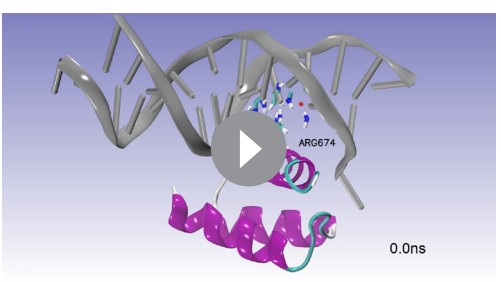

**Video 3.** The 350-ns molecular dynamics (MD) simulation trajectory of 1HF0-based HB4–dsDNA complex. The HB4–dsDNA complex is in secondary structure cartoon representation (violet color: HB4's helix; gray color: dsDNA). The top C-terminal helix residue that most contacts the DNA, which is ARG674, and the DNA atoms within 4 Å distance from it are in licorice representation.

https://elifesciences.org/articles/70412/figures#video3

## Discussion

In this study, we revealed that *ZHX2* is an important oncogene through mediating a novel molecular mechanism in TNBC (*Figure 6J*). Depletion of ZHX2 leads to decreased TNBC cell proliferation as well as invasion. By performing gene expression analyses, ZHX2-regulated genes display a

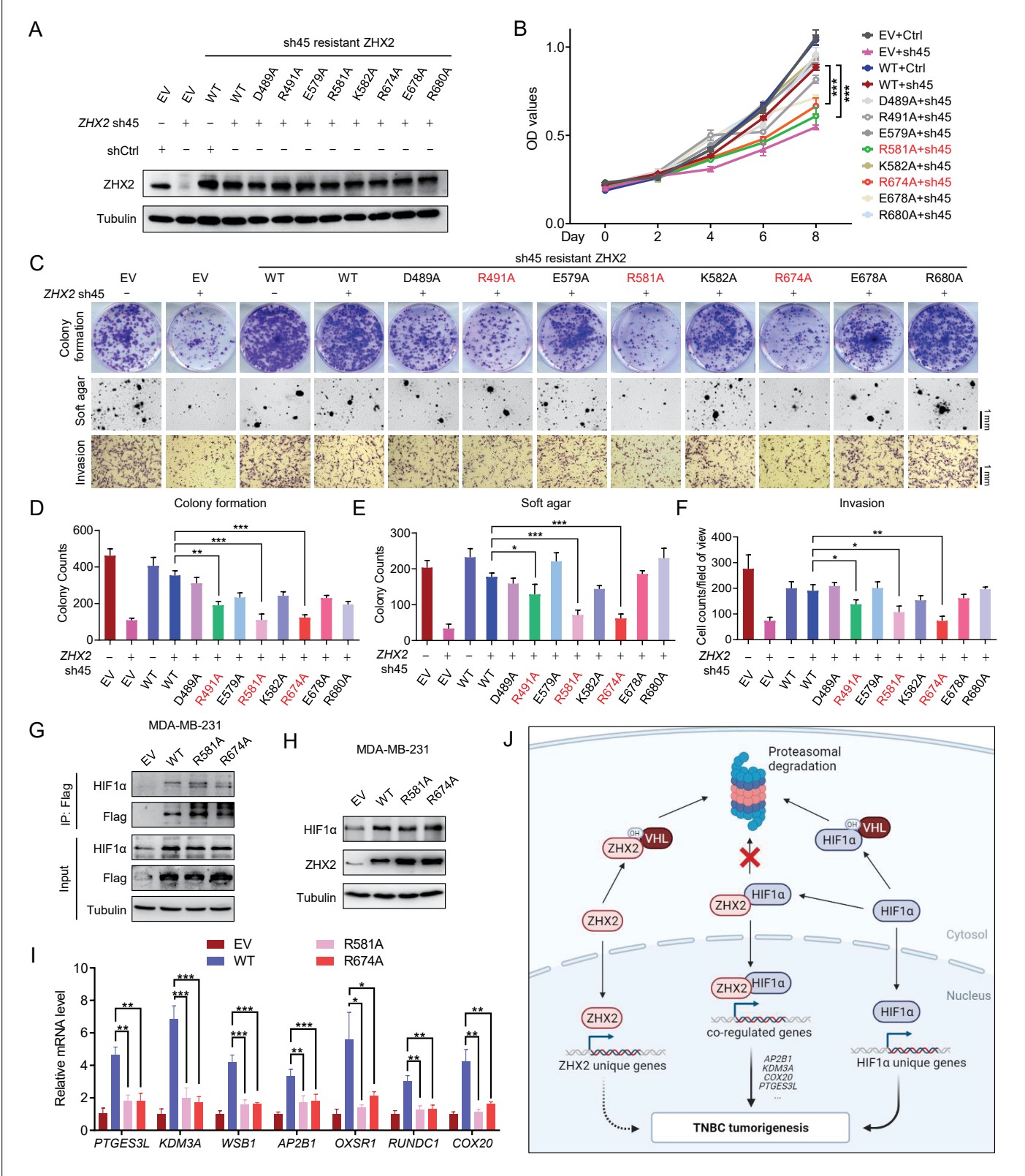

**Figure 6.** Identification of important sites on ZHX2 that may affect its function in triple-negative breast cancer (TNBC). (**A–F**) Immunoblot of cell lysates (**A**), cell proliferation (**B**), representative images of 2D colony formation (top), 3D soft agar (middle), and invasion assay (bottom) (**C**) as well as the corresponding quantifications (**D–F**) of MDA-MB-231 cell lines infected with lentivirus encoding either sh45-resistant *ZHX2* wild type (WT) or mutations, followed by sh45 *ZHX2* depletion. (**G–I**) Immunoprecipitations (**G**), immunoblot (**H**), and qRT-PCR (**I**) of MDA-MB-231 cells infected with lentivirus

*Figure 6 continued on next page*

*Figure 6 continued*

encoding either ZHX2 WT or mutation. (**J**) Schematic model of the major findings from the current study. Error bars represent mean ± standard error of the mean (SEM), unpaired *t*-test. *p < 0.05; **p < 0.01; ***p < 0.001.

The online version of this article includes the following figure supplement(s) for figure 6:

**Source data 1.** Uncropped western blot images for *Figure 6*.

**Figure supplement 1.** Residue-based DNA contact analysis derived from ensembles of DNA-bound HD proteins.

significant overlap with HIF-regulated genes in TNBC. ZHX2 binds with HIF1α and HIF1β and regulates HIF1α protein levels and transcriptional activity. By using structural simulation and the reconstitution system, we pinpoint residues (R491, R581, and R674) on ZHX2 that may be important for its DNA-binding function as well as tumorigenic potential. Overall, our study establishes an important role of ZHX2 in regulating HIF1α signaling and tumorigenesis in TNBC.

From a genomic perspective, it is known that *ZHX2* is located on 8q24, a chromosomal region frequently amplified in cancers. Indeed, *ZHX2* is amplified in several cancers, including breast cancer, ovarian cancer, and prostate cancer. In most cases, *ZHX2* is coamplified with another well-established oncogene *MYC* (*Figure 1A–C*). This finding bears several implications. First, it suggests that ZHX2 and c-Myc may act in concert in promoting tumorigenesis. Second, the role of ZHX2 in cancers can be context dependent. Although ZHX2 may be amplified in multiple cancers, its protein levels can be regulated post-transcriptionally. Our previous research showed that in ccRCC, ZHX2 can be regulated by pVHL potentially through hydroxylation on multiple proline residues in the PRR domain (*Zhang et al., 2018*). Therefore, the presence or absence of factors that mediate prolyl hydroxylation in the same niche as ZHX2 can dictate its regulation and downstream function. Further, it remains uncertain whether ZHX2 interacts with DNA directly or indirectly via other transcription factors to exert its transcriptional regulation on downstream target genes. The repertoire of different coactivators/repressors with which ZHX2 may interact can therefore govern its localization in the genome and thus its downstream function in different cancer settings.

By comparing the RNA-seq and ChIP-seq results, we identified seven interesting candidate genes which might be essential for maintaining the oncogenic function of ZHX2. Further rescue experiments showed at least four genes (*AP2B1*, *COX20*, *KDM3A*, and *PTGES3L*) contributed to the oncogenic role of ZHX2 in an accumulative fashion (*Figure 5F*). AP2B1 is a component of the adaptor protein complex 2 (AP-2) (*Lau and Chou, 2008*). This gene encodes a member of the AP-2 family of transcription factors. AP-2 proteins form homo- or heterodimers with other AP-2 family members and bind specific DNA sequences. They are thought to stimulate cell proliferation and suppress terminal differentiation of specific cell types during embryonic development. COX20 (also known as FAM36A; MIM#614698) is a mitochondrial inner membrane protein, whose known function is to chaperone COX2, a subunit of cytochrome c oxidase in the yeast's mitochondrial matrix (*Hell et al., 2000*; *Otero et al., 2019*). It contains two transmembrane helices and localizes to the mitochondrial membrane. Mutations in this gene can cause mitochondrial complex IV deficiency, which results in ataxia and muscle hypotonia (*Keerthiraju et al., 2019*; *Otero et al., 2019*). KDM3A (Lysine Demethylase 3A) is a zinc finger protein that contains a jumonji domain and had identified as the histone H3 lysine nine mono- and dimethyl demethylase enzyme (*Wang et al., 2013*). KDM3A is overexpressed in breast cancer tissues and our previous siRNA screening indicated that it may be important for TNBC cell growth (*Liao et al., 2020*; *Yoo et al., 2020*). PTGES3L (Prostaglandin E Synthase 3-Like) is predominantly expressed in skeletal muscle. Diseases associated with PTGES3L include arthrogryposis, distal, type 2A, and spondylocarpotarsal synostosis syndrome. Limited research has explored the function of PTGES3L in tumors. Although we have identified aforementioned ZHX2 downstream targets, it is worth mentioning that ZHX2 may employ other HIF independent and unique downstream targets for its oncogenic function in TNBC (*Figure 6J*).

Thus far, it remains unclear which residues on ZHX2 are critical for its transcriptional activity as well as its oncogenic role in cancer. We also did MD simulation to confirm the import residues for regulating the phenotype of ZHX2. We found that the top DNA-contacting residues across the three HDs were arginine namely Arg491, Arg581, and Arg674. In terms of interaction, it can be observed that Arg491 and Arg581 of HD2 and 3, respectively, were seen to be interacting with the DNA's phosphate backbone throughout the 350 ns MD simulations (*Videos 1 and 2*). However, in the case of HD 4,

Arg674 was seen to be interacting first with the nucleobases within 4 Å distance. Shortly after ~120 ns, the DNA slowly drifted away but this was prevented because of an interaction between Arg674 and the DNA's phosphate backbone (*Video 3*). These arginine–DNA phosphate backbone interactions could be critical for the stabilization of transcription factor binding with the DNA to either stimulate or repress the transcription of a specific gene. By performing ZHX2 depletion and reconstitution experiments, we found that several residues located in the HDs of ZHX2 may be important for its oncogenic role in TNBC (*Figure 6A–F*). In addition, these three arginine residues are found to be important on mediating the effect of ZHX2 on transactivating HIF1α activity in TNBC cells (*Figure 6I*). Further research needs to be performed to determine whether these residues are critical for ZHX2 localization to DNA, either directly or indirectly via recruitment of coactivators/repressors. Lastly, given that we already found three residues important for the function of ZHX2 in TNBC, we can potentially design small peptides to competitively bind ZHX2 and inhibit its localization to DNA. By engineering these peptides to be membrane permeable, we can potentially test whether they can inhibit the oncogenic role of ZHX2 in TNBC. Given that ZHX2 inhibitors are still not available at this time, these peptide inhibitors can be used as a proof-of-principle approach to motivate further development of specific ZHX2 inhibitors in potential cancer therapies.

However, our study does carry some limitations. For examples, the detailed mechanism by which ZHX2 leads to HIF1α protein stability needs to be further examined. In addition, further investigation will be needed to perform on how to therapeutically target ZHX2 oncogenic signaling by repressing its transcriptional activity in TNBC.

Despite these limitations, our study investigates how ZHX2 signaling may transactivate HIF activity in TNBC, therefore contributing to breast tumorigenesis. In addition, we elucidate some of potential sites on ZHX2 that may be important for its oncogenic function in TNBC.

## Materials and methods

### Key resources table

| Reagent type (species) or resource | Designation | Source or reference | Identifiers | Additional information |
|---|---|---|---|---|
| Gene (*Homo sapiens*) | *ZHX2* | GenBank | Gene ID: 22,882 | |
| Cell line (*Homo sapiens*) | MDA-MB-231 | ATCC | Catalog # ATCC HTB-26 RRID:CVCL_0062 | |
| Cell line (*Homo sapiens*) | MDA-MB-468 | ATCC | Catalog # ATCC HTB-132 RRID:CVCL_0419 | |
| Antibody | Rabbit anti ZHX2 antibody (Rabbit Polyclonal) | Genetex | Catalog # 112,232 RRID:AB_10727176 | WB (1:1000) Co-IP (1:100) IHC (1:500) |
| Antibody | Rabbit anti HIF1 α (Rabbit Monoclonal) | Cell Signaling | Catalog # 36,169 RRID:AB_2861751 | WB (1:1000) |
| Antibody | Rabbit anti VHL (Rabbit Polyclonal) | Cell Signaling | Catalog # 68,547 | WB (1:1000) IHC (1:200) |
| Recombinant DNA reagent | pLKO.1 | Addgene | Catalog # 10,878 RRID:Addgene_10878 | TRC cloning vector |
| Recombinant DNA reagent | lentiCRISPR v2 | Addgene | Catalog # 52,961 RRID:Addgene_52961 | |
| Recombinant DNA reagent | pcDNA3.1 | Addgene | Catalog # 52,535 RRID:Addgene_52535 | |
| Chemical compound, drug | MG132 | Peptide International | Catalog # IZL-3175-v | |
| Chemical compound, drug | Doxycycline hydrochloride | Sigma-Aldrich | Catalog # D3447 | |
| Chemical compound, drug | Agarose | Life Technologies | Catalog # BP165-25 | Soft agar assay |
| Commercial assay or kit | MTS Assay Kit (Colorimetric) | Abcam | Catalog # ab197010 | Cell proliferation assay |
| Commercial assay or kit | BD BioCoat Matrigel Invasion Chamber | Corning | Catalog # ab354480 | Invasion assay |

## Cell culture and reagents

MDA-MB-231, MDA-MB-436, MCF-7, Hs578T, and 293T cells were cultured in Dulbecco's modified Eagle's medium (DMEM) (GIBCO 11965118) supplemented with 10% fetal bovine serum (FBS) and 1 % penicillin–streptomycin (Pen Strep). T47D, BT474, HCC1428, HCC3153, HCC1143, HCC70, and MDA-MB-468 cells were cultured in 10% FBS, 1% Pen Strep RPMI 1640 (GIBCO 11875093). Normal breast epithelial cells HMLE and MCF-10A were cultured in MEGM (Lonza CC-3151) containing SingleQuots Supplements (Lonza CC-4136). 293T cells were obtained from UNC Tissue Culture Facility. HMLE cell line was provided by Dr. Wenjun Guo at Albert Einsetein College of Medicine and HCC3153 was from UT Southwestern Cell Line core. All other cell lines were obtained from ATCC. All cells were authenticated via short tandem repeat testing. Mycoplasma detection was routinely performed to ensure cells were not infected with mycoplasma by using MycoAlert Detection kit (Lonza, LT07-218). Cells were maintained at 37°C in a 5% $CO_2$ incubator. Cells were incubated overnight in 1% $O_2$ hypoxia chamber for hypoxia treatment. Dox (D9891) was purchased from Sigma-Aldrich, DMOG (D1070-1g) was from Frontier Scientific, and MG132 (IZL-3175-v) was from Peptide International.

## Antibodies

Antibodies used for immunoblotting, IP, and IHC staining were as follows: Rabbit anti ZHX2 antibody (Genetex, 112232), Rabbit anti HIF1α (Cell Signaling, 36169), Rabbit anti HIF1β (Cell Signaling, 5537), Rabbit anti VHL (Cell Signaling, 68547), rabbit anti HA tag (Cell Signaling, 3724), mouse anti α-Tubulin (Cell Signaling, 3873), mouse anti Ubiquitin (Santa Cruz, sc-8017), and Rabbit anti NF-κB p65 (Cell Signaling, 8242S). Peroxidase conjugated goat anti-mouse secondary antibody (31430) and peroxidase conjugated goat anti-rabbit secondary antibody (31460) were from Thermo Scientific.

## Plasmids

pBABE HA-*VHL*, pcDNA-3.1-FLAG-HA-*ZHX2* (WT), pcDNA-3.1-FLAG-HA-*ZHX2* (*ZHX2* sh45 resistant), and pcDNA-3.1-HA-*HIF1A* were previously described. pcDNA-3.1-FLAG-HA-*ZHX2* (D489A), pcDNA-3.1-FLAG-HA-*ZHX2* (R491A), pcDNA-3.1-FLAG-HA-*ZHX2* (E579A), pcDNA-3.1-FLAG-HA-*ZHX2* (R581A), pcDNA-3.1-FLAG-HA-*ZHX2* (K582A), pcDNA-3.1-FLAG-HA-*ZHX2* (R674A), pcDNA-3.1-FLAG-HA-*ZHX2* (E678A), and pcDNA-3.1-FLAG-HA-*ZHX2* (R680A) were constructed using standard molecular biology techniques. Quick Change XL Site-Directed Mutagenesis Kit (200516, Agilent Technologies) was used to construct *ZHX2* mutants. The GATEWAY Cloning Technology (11789020 and 11791019, Invitrogen) was used to recombine plasmids for virus production. All plasmids were sequenced to confirm validity.

## Lentiviral shRNA and sgRNA vectors

Lentiviral *ZHX2* shRNAs (pLKO vector based) were obtained from Broad Institute TRC shRNA library. sgRNAs were cloned into the lentiCRISPR v2 backbone (Addgene, Plasmid #52961). Target sequences were as follows:

> Control shRNA: AACAGTCGCGTTTGCGACTGG
> *ZHX2* shRNA (*43*): CCCACTAAATACTACCAAATA
> *ZHX2* shRNA (*45*): CCGTAGCAAGGAAAGCAACAA
> *HIF1A* shRNA (*3809*): CCAGTTATGATTGTGAAGTTA
> *HIF1A* shRNA (*3810*): GTGATGAAAGAATTACCGAAT
> Control sgRNA: GCGAGGTATTCGGCTCCGCG
> *VHL* sgRNA (*1*): CATACGGGCAGCACGACGCG
> *VHL* sgRNA (*2*): GCGATTGCAGAAGATGACCT
> *VHL* sgRNA (*8*): ACCGAGCGCAGCACGGGCCG

## Survival analysis

The K–M plots were got from https://kmplot.com (*Györffy et al., 2010*). We chose TNBC patients as follows: ER status IHC: ER negative, ER status array: ER negative, PR status IHC: PR negative, and HER2 status array: HER2 negative. Finally, 153 TNBC patients were included in the overall survival analysis. ZHX2 overexpression were chosen as upper tertile expression.

## Amplification status of *ZHX2* in different breast cancer subtype

All data in *Supplementary file 1a* were got from cBioPortal (https://www.cbioportal.org/; *Gao et al., 2013*). We searched the percentage of ZHX2 amplification in all breast cancer datasets, and

found ZHX2 was mainly amplified in seven datasets: Breast Cancer (METABRIC, Nature 2012, *Curtis et al., 2012* and Nat Commun 2016, *Pereira et al., 2016*), The Metastatic Breast Cancer Project (Provisional, February 2020), Breast Invasive Carcinoma (TCGA, Cell 2015; *Ciriello et al., 2015*), Breast Invasive Carcinoma (TCGA, Firehose Legacy), Breast Invasive Carcinoma (TCGA, PanCancer Atlas), Metastatic Breast Cancer (INSERM, PLoS Med 2016; *Lefebvre et al., 2016*), and Breast Invasive Carcinoma (TCGA, Nature 2012; *Cancer Genome Atlas Network, 2012*). Two datasets, Breast Invasive Carcinoma (TCGA, PanCancer Atlas) and Metastatic Breast Cancer (INSERM, PLoS Med 2016; *Lefebvre et al., 2016*), did not show the ER, PR, and HER2 status, and were excluded from our study. In all the datasets, ER and PR status were determined by IHC. Three datasets, Breast Cancer (METABRIC, Nature 2012, *Curtis et al., 2012* and Nat Commun 2016, *Pereira et al., 2016*), The Metastatic Breast Cancer Project (Provisional, February 2020), and Breast Invasive Carcinoma (TCGA, Cell 2015; *Ciriello et al., 2015*) assigned the HER2 status by the original researchers. Two datasets, Breast Invasive Carcinoma (TCGA, Cell 2015; *Ciriello et al., 2015*) and Breast Invasive Carcinoma (TCGA, Firehose Legacy) assigned the HER2 status by two standard, IHC and fluorescence in situ hybridization (FISH). In this study, HER2+ status in these two datasets, were determined by IHC.

## Virus production and infection

293T packaging cell lines were used for lentiviral amplification. Lentiviral infection was carried out as previously described (*Zhang et al., 2018*). Briefly, viruses were collected at 48 and 72 hr post-transfection. After passing through 0.45 μm filters, viruses were used to infect target cells in the presence of 8 μg/ml polybrene. Subsequently, target cell lines underwent appropriate antibiotic selection. MDA-MB-231/468 cells infected with lentivirus encoding shRNAs or sgRNAs were selected by puromycin (1 μg/ml) for 48 hr. MDA-MB-231/468 cells infected with lentivirus encoding *ZHX2* wild type or mutation were selected by hygromycin (50 μg/ml) for 72 h.

## Immunoblotting and IP

EBC buffer (50 mM Tris–HCl pH8.0, 120 mM NaCl, 0.5% NP40, 0.1 mM EDTA, and 10% glycerol) supplemented with complete protease inhibitor and phosphoSTOP tablets (Roche Applied Bioscience) was used to harvest whole cell lysates at 4°C. Cell lysate concentrations were measured by Protein assay dye (BioRad). An equal amount of cell lysates was resolved by SDS–PAGE. For IP, whole-cell lysates were prepared in EBC buffer supplemented with protease inhibitor and phosphatase inhibitor. The lysates were clarified by centrifugation and then incubated with primary antibodies or HA antibody conjugated beads (HA beads, Roche Applied Bioscience) overnight at 4°C. For primary antibody incubation, cell lysates were incubated further with protein G sepharose beads (Roche Applied Bioscience) for 2 hr at 4°C. The bound complexes were washed with EBC buffer 5× times and were eluted by boiling in SDS loading buffer. Bound proteins were resolved in SDS–PAGE followed by immunoblotting analysis.

## Human samples and IHC

The 20 pairs of normal/tumor TNBC patient tissue used for western blotting were obtained from tissue management core facility from UNC-Chapel Hill (UNC approval number, 15-0041). The 20 pairs of normal/tumor TNBC patient parafilm tissue slides used for IHC were obtained from tissue management core facility from University of Texas Southwestern Medical Center (UTSW reference number, Y1-21-733). The TNBC TMA were bought from US Biomax (BR1301). The current study was approved by the Institutional Ethics Committee of University of Texas Southwestern Medical Center (the IRB exemption approval letter will be provided upon request). Deparaffinized sections were subjected to antigen retrieval in 0.01 M citrate buffer solution. After blocking of endogenous peroxidase activity in 3% $H_2O_2$, the sections were incubated with the ZHX2 antibody at the dilution of 1:500 or pVHL antibody at the dilution of 1:200 overnight at 4°C. Then the sections were rinsed and visualized by immunoperoxidase staining with the Real Envision Detection Kit according to the manufacturer's instructions or detected by immunofluorescence analysis. The IHC localization was scored in a semi-quantitative fashion incorporating both the intensity and distribution of specific staining by using H score (*Detre et al., 1995*).

## Luciferase reporter assay

For HIF transcription assay, subconfluent MDA-MB-231 cells (200,000 cells/24-well plate) were transiently transfected with 30 ng pCMV-Renilla and100 ng of HRE-Luci reporter. In order to text the relation between ZHX2 concentration and HIF reporter activity, subconfluent HEK293T cells (200,000 cells/24-well plate) were transiently transfected with different amount of HA-ZHX2 (100, 200, and 400 ng), 100 ng HRE-Luci reporter and 30 ng pCMV-Renilla. Forty-eight hours after transfection, luciferase assays were performed by Dual-Luciferase Reporter Assay System (Promega, E1960). The experiments were repeated in triplicate with similar results.

## Construction of shRNA-resistant *ZHX2*^res cell lines

pcDNA-3.1-FLAG-HA-*ZHX2* (WT) were used as template. The target region of *ZHX2* shRNA (*Zhang et al., 2018*) was the nucleotide 2120–2140: CCGTAGCAAGGAAAGCAACAA. We mutated 2124 A to T, 2127 A to T, 2130 A to T, 2135 A to T by using Quick Change XL Site-Directed Mutagenesis Kit (200516, Agilent Technologies). This mutated plasmid was resistant to *ZHX2* shRNA (*Zhang et al., 2018*). The GATEWAY Cloning Technology (11789020 and 11791019, Invitrogen) was used to recombine plasmids for virus production. All plasmids were sequenced to confirm validity. MDA-MB-231/468 cells were firstly infected with lentivirus encoding *ZHX2* wild type for 48 hr, and then selected by hygromycin (50 µg/ml) for 72 hr. Afterwards, MDA-MB-231/468 cells were infected with lentivirus encoding *ZHX2* sh45 for 48 hr and selected by puromycin (2 µg/ml) for 72 hr.

## Construction of cell lines for the rescue experiment

pLenti6.3-based expression constructs (*PTGES3L*, *KDM3A*, *WSB1*, *AP2B1*, *OXSR1*, and *COX20*) were bought from DNASU (https://dnasu.org/DNASU/Home.do). We verified these plasmids by sequencing before use. Upon generating stable blasticidin selected cell lines for each gene in MDA-MB-231, we subsequently knocked down *ZHX2* by sh45 in each stable cell line and selected by puromycin for 4 days followed by performing phenotype studies.

## 2D cell proliferation assay

For MTS assay, cells were seeded in triplicate in 96-well plates (1000 cells/well) in appropriate growth medium. At indicated time points, cells were replaced with 90 µl fresh growth medium supplemented with 10 µl MTS reagents (Abcam, ab197010), followed by incubation at 37°C for 1–4 hr. OD absorbance values were measured at 490 nm using a 96-well plate reader (BioTek). For colony formation assays, cells were seeded in duplicate in 6-well plates ($2 \times 10^3$ cells/well) in appropriate growth medium. Media was changed every 2 days. After 7 days, cells were fixed with 4% formaldehyde for 10 min at room temperature, stained for 10 min with 0.5% crystal violet, and then washed several times with distilled water. Once dried, the plates were scanned.

## 3D anchorage-independent soft agar assay

Cells were plated in a top layer at a density of 10,000 cells/ml in complete medium with 0.4% agarose (Life Technologies, BP165-25), onto bottom layers composed of medium with 1% agarose followed by incubation at 4°C for 10 min. Afterwards, cells were moved to a 37°C incubator. Every 4 days, 200 µl of complete media were added onto the plate. After 2–4 weeks, the extra liquid on the plate was aspirated, and 1 ml medium supplemented with 100 µg/ml iodonitrotetrazoliuim chloride solution was added onto each well. After incubating overnight at 37°C, the colonies were captured by an image microscope and quantified after a whole plate scan.

## Cell invasion assay

MDA-MB-231 and MDA-MB-468 cell invasion assay was performed using BD BioCoat Matrigel Invasion Chamber (354480) according to the manufacturer's instructions. In total, $3 \times 10^4$ (for MDA-MB-231) and $3 \times 10^5$ (for MDA-MB-468) cells were inoculated into each chamber in triplicate and incubated for 18 hr at 37°C, 5% $CO_2$ incubator. The cells on the lower surface of the membrane were stained using Diff-Quick stain kit (B4132-1A) from SIEMENS, and then counted under EVOS XL Core Microscope (Cat# AMEX1000, Thermo Fisher Scientific).

## RNA-seq analysis

Procedures was described previously (*Liao et al., 2020*). Briefly, total RNA from triplicates was extracted from MDA-MB-231 cells infected with control or *ZHX2* shRNAs by using RNeasy kit with on column DNase digestion (Qiagen). Library preparation and sequencing were performed by BGI as paired end 50 bp reads. Reads were then filtered for adapter contamination using cutadapt (*Patro et al., 2017*) and filtered such that at least 90% of bases of each read had a quality score >20. Reads were aligned to the reference genome (hg19) using STAR version 2.5.2b, and only primary alignments were retained (*Love et al., 2014*). Reads overlapping blacklisted regions of the genome were then removed. Transcript abundance was then estimated using salmon (*Miller et al., 2012*), and differential expression was detected using DESeq2 (*Bird et al., 2010*). RNA-seq data are available at GSE175487. Pathway enrichments were calculated using g:Profiler (*Reimand et al., 2019*) where the pathway database was supplemented with the list of HIF DKO downregulated genes to obtain the adjusted *p* value indicating a significant association. Association with the HALLMARK hypoxia pathway was conducted using GSEA (*Reimand et al., 2019*).

## Real-time PCR

Total RNA was isolated with RNeasy mini kit (Qiagen). First strand cDNA was generated with an iScript cDNA synthesis kit (BioRad). Real-time PCR was performed in triplicate. Real-time PCR primer sequences are listed in *Supplementary file 1f*.

## ChIP-seq analyses

MDA-MB-231 cells were infected by HA-ZHX2 which is resistant to *ZHX2* sh45, and then infected by ZHX2 sh45. ChIP was performed with HA tag (Cell Signaling, 3724). The ChIP-seq library was prepared using a ChIP-seq DNA sample preparation kit (Illumina) according to the manufacturer's instructions. Samples were sequenced on an Illumina HiSeq2500 with single-end 76 bp reads. Reads were then filtered for adaptor contamination using Cutadapt and filtered such that at least 90% of bases of each read had a quality score >20. Duplicated sequences were then capped at a maximum of five occurrences, and reads were aligned to the reference genome (hg19) using STAR (*Dobin et al., 2013*) version 2.5.2b retaining only primary alignments. Reads overlapping blacklisted regions of the genome were then removed. Reads were then extended in silico to a fragment size of 250 bp, and regions of significant enrichment relative to input control were identified using MACS2 (*Zhang et al., 2008*). A unified set of enriched regions for ZHX2 was obtained by taking the intersection of the two replicates using bedtools (*Quinlan and Hall, 2010*). ChIP-seq data for HIF1α were obtained from GSE108833, and data for H3K4me3 and H3K27ac were obtained from GSE49651. ChIP enrichment heatmaps over promoters were generated using deepTools (*Ramírez et al., 2016*).

## Orthotopic tumor xenograft

Procedures for animal studies were described previously (*Liao et al., 2020*). Briefly, 6-week-old female NSG mice (Jackson lab) were used for xenograft studies. Approximately $1 \times 10^6$ viable MDA-MB-231-luc or HCC70-luc cells expressing Teton control or Teton *ZHX2* shRNAs were resuspended in 1: 1 ratio in 50 μl medium and 50 μl matrigel (Corning, 354234) and injected orthotopically into the fourth mammary fat pad of each mouse. After cell injection and following two consecutive weeks of tumor monitoring to ensure the tumor was successfully implanted, mice were fed Purina rodent chow with Dox (Research Diets Inc, #5001). Tumor size was measured twice a week using an electronic caliper. Tumor volumes were calculated with the formula: volume = $(L \times W^2)/2$, where *L* is the tumor length and *W* is the tumor width measured in millimeters. The rough mass of tumors was presented as mean ± standard error of the mean SEM and evaluated statistically using *t*-test. After mice were sacrificed, lung ex vivo imaging was performed immediately to examine tumor metastasis. All animal experiments were in compliance with National Institutes of Health guidelines and were approved by the University of Texas, Southwestern Medical Center Institutional Animal Care and Use Committee.

## Binding free energy calculations using MM/PBSA

To calculate the binding Gibbs free energy change between two groups of molecules, we used MM/PBSA continuum solvation approach (*Miller et al., 2012*) to analyze trajectories out of MD simulations (detailed in Supporting Information). In this approach, the binding Gibbs free energy change, $\Delta G_{bind}$,

can be expressed in terms of the *xyz* (ΔGMM), *xyz* (ΔGsolv), and the entropy of the system (*T*Δ*S*) as shown in *Equation 1*.

$$\Delta G_{\text{bind}} = \Delta G_{\text{MM}} + \Delta G_{\text{solv}} - T\Delta S \qquad (1)$$

where $\Delta G_{\text{MM}} = \Delta G_{\text{elec}} + \Delta G_{\text{VDW}}$ and $\Delta G_{\text{solv}} = \Delta G_{\text{polar}} + \Delta G_{\text{nonpolar.}}$

## DNA–protein contact analysis from structural bioinformatics data

Sequences in the apo-form human HDs 2, 3, and 4 (HD2/3/4) (PDB ID: 3NAU, 2DMP, and 3NAR) (*Bird et al., 2010*; *Ohnishi et al., 2020*) were BLASTed against the sequences in the Protein Data Bank (PDB; *Berman et al., 2000*). Among the resolved structures, 12, 20, and 15 DNA-bound complexes with sequence identity higher than 30% (*Rost, 1999*) for HD2, HD3, and HD4 were identified. We then carried out the homology modeling, using SWISS-MODEL (*Waterhouse et al., 2018*), to structurally model the HD2/3/4 using their corresponding bound forms as the templates, so that their sequences assume the protein structures in the DNA-complexed forms. For instance, HD2 sequence could therefore adapt 12 different bound-form protein structures. With this method, we found every HD protein contact the DNA with its last (C-terminal) helix (*Figure 6—figure supplement 1*). We then count the number of DNA–protein contacts at the atomic level for every residue in the C-terminal helix. The top-ranked residues in HD2/3/4, in terms of their DNA contact frequency normalized by the number of bound forms used, are listed together with their evolutionary conservation in *Supplementary file 1*.

## MD simulations

### Homology modeling to create DNA-bound HD complexes for simulations

Because there are no experimentally solved DNA-complexed structures for human HD2, 3, and 4, in order to simulate the human HD–DNA interaction, our goal is to find structurally solved DNA-bound forms whose DNA sequence could have the highest chance to stably interact with human HD2/3/4. We aimed to ensure the highest likelihood of stable interaction between the selected DNA and HD2/3/4 as well as to have a fair comparison of binding ability for HD2/3/4 and their DNA-binding residues. To this end, we searched the DNA-bound HD proteins containing a DNA-binding helical stretch that has the highest sequence identity with the C-terminal helices (the main DNA-binding helix; see *Figure 6—figure supplement 1* and *Video 1*) in HD2, 3, and 4, respectively. Among the bound-form proteins that have the top two highest sequence homology in the C-terminal helices with those in human HD2/3/4, by homology modeling using SWISS-MODEL (*Waterhouse et al., 2018*), a NMR-resolved structural ensemble of VND/NK-2 HD–DNA complex (PDB ID: 1NK2, where the first model is taken) (*Gruschus et al., 1997*) was chosen to build the DNA-bound form of HD2, an X-ray-resolved structure of Yeast MATα2 HD/MCM1 transcription factor/DNA complex (PDB ID: 1MNM; *Tan and Richmond, 1998*) was chosen to build the DNA-bound form of HD3, and an X-ray-resolved structure of Oct-1 Transcription factor DNA complex (PDB ID:1HF0; *Reményi et al., 2001*) was chosen to build the DNA-bound form of HD4, respectively. The homology of C-terminal helices between HD2/3/4 and their corresponding bound-form templates are 54.55%, 53.85%, and 100%, respectively.

### System setup and energy minimization

Prior to solvation and addition of ions, protonation state and the net charge of HD2/3/4–dsDNA complexes at pH 7.0 were calculated using PDB2PQR (*Dolinsky et al., 2007*). The starting structure was prepared using ff14SB (*Maier et al., 2015*) force fields for proteins, bsc1 (*Ivani et al., 2016*) force fields for the DNA, TIP3P water model, and monovalent ion parameters (*Joung, 2009*) through tLeap (*Daoudi et al., 2019*) from AmberTools18. To neutralize the charge of each system, 24 Na$^+$ were added into hb2–dsDNA and hb3–dsDNA complexes and 23 Na$^+$ were added into hb4–dsDNA complex systems. In addition, 23 Na$^+$ and 23 Cl$^-$ ions were added to each system to reach 100 mM salt concentration. Each system was prepared in a water box measuring 78 Å on all sides.

Energy minimization for each of the systems was done in two stages. In the first stage, a harmonic restraint of 100 kcal/mol/Å$^2$ was applied on all heavy atoms of both protein and dsDNA. In the second stage, the harmonic restraints for protein's CA atoms were relaxed to 2 kcal/mol/Å$^2$ while all the DNA's heavy atoms were still subject to a 100 kcal/mol/Å$^2$ restraint.

## Equilibration and explicit solvent production MD simulations

Each energy-minimized system was gradually heated from 50 to 320 K and cooled down to 310 K in a canonical (NVT) ensemble, using Langevin thermostat (*Pastor et al., 2006*) with a collision frequency of 2 ps$^{-1}$, for 25 ps while applying harmonic restraints of 10 kcal/mol/Å$^2$ on dsDNA's C2, C4', and P atoms and 2 kcal/mol/Å$^2$ on protein's CA atoms. Each of the systems was equilibrated first in a canonical ensemble at 310 K for 15 ns. This was followed by an isothermal–isobaric ensemble for 20 ns at 310 K applying harmonic restraints of 2 kcal/mol/Å$^2$ on dsDNA's C2, C4', and P atoms and 1 kcal/mol/Å$^2$ on protein's CA atoms. Further equilibration isothermal–isobaric ensemble Particles are simulated with constant Number, Pressure and Temperature (NPT), where a constant pressure was maintained by Berendsen barostat (*Berendsen et al., 1984*) at 1 atm and 310 K, was done for 40 ns, while harmonic restraints of 2 kcal/mol/Å$^2$ on dsDNA's C2, C4', and P atoms and 0.1 kcal/mol/Å$^2$ on protein's CA atoms were applied. This was followed by a 350 ns production run at 2 fs time step applying the SHAKE constraint algorithm (*Hopkins et al., 2015*) to hydrogen atoms in isothermal–isobaric ensemble at 310 K and 1 atm. All the simulations were carried out by the AMBER18 software package (*Daoudi et al., 2019*) with long-range electrostatic forces being calculated using Particle Mesh Ewald method (*Darden et al., 1993*) at a 10 Å cutoff distance.

## Statistical analysis

All statistical analysis was conducted using Prism 8.0 (GraphPad Software). All graphs depict mean ± SEM unless otherwise indicated. Statistical significances are denoted as n.s. (not significant; $p > 0.05$), $*p < 0.05$, $**p < 0.01$, $***p < 0.001$, $****p < 0.0001$. The numbers of experiments are noted in figure legends. To assess the statistical significance of a difference between two conditions, we used unpaired two-tail Student's *t*-test. For experiments comparing more than two conditions, differences were tested by a one-way ANOVA followed by Dunnett's or Tukey's multiple comparison tests.

## Acknowledgements

This work was supported in part by the National Cancer Institute (Q.Z., R01CA211732 and R21CA256833), Cancer Prevention and Research Institute of Texas (CPRIT, RR190058 to Q.Z.), and American Cancer Society (RSG-18-059-01-TBE). J.M.S. and T.S.P. were supported by NINDS (P30NS045892). Q.Z. is an American Cancer Society Research Scholar, CPRIT Scholar in Cancer Research, V Scholar, Kimmel Scholar, Susan G Komen Career Catalyst awardee, and Mary Kay Foundation awardee.

## Additional information

### Competing interests

Yan Peng: The other authors declare that no competing interests exist.

### Funding

| Funder | Grant reference number | Author |
|---|---|---|
| National Cancer Institute | R01CA211732 | Qing Zhang |
| National Cancer Institute | R01CA256833 | Qing Zhang |
| Cancer Prevention and Research Institute of Texas | RR190058 | Qing Zhang |
| American Cancer Society | RSG-18-059-01-TBE | Qing Zhang |
| National Institute of Neurological Disorders and Stroke | P30NS045892 | Jeremy M Simon Travis S Ptacek |
| American Cancer Society | Research Scholar | Qing Zhang |
| Cancer Prevention and Research Institute of Texas | Scholar in Cancer Research | Qing Zhang |

| Funder | Grant reference number | Author |
|---|---|---|
| V Foundation | V Scholar | Qing Zhang |
| Sidney Kimmel Foundation | Kimmel Scholar | Qing Zhang |
| Susan G Komen Breast Cancer FDN, Inc | Career Catalyst Award | Qing Zhang |
| Mary Kay Foundation | | Qing Zhang |

The funders had no role in study design, data collection, and interpretation, or the decision to submit the work for publication.

## Author contributions

Wentong Fang, Formal analysis, Investigation, Validation, Writing - original draft; Chengheng Liao, Conceptualization, Data curation, Formal analysis, Investigation, Methodology, Project administration, Resources, Supervision, Validation, Visualization, Writing - original draft, Writing - review and editing; Rachel Shi, Travis S Ptacek, Formal analysis, Investigation; Jeremy M Simon, Christopher Llynard Ortiz, Software; Giada Zurlo, Ujjawal Manocha, Methodology; Youqiong Ye, Leng Han, Cheng Fan, Investigation; Lei Bao, Weibo Luo, Resources; Hong-Rui Lin, Methodology, Software; Yan Peng, Data curation, Formal analysis, Investigation; William Y Kim, Resources, Supervision; Lee-Wei Yang, Supervision; Qing Zhang, Conceptualization, Funding acquisition, Project administration, Resources, Writing - original draft, Writing - review and editing

## Author ORCIDs

Wentong Fang (iD) http://orcid.org/0000-0003-0047-1198
Chengheng Liao (iD) http://orcid.org/0000-0002-9073-3835
Jeremy M Simon (iD) http://orcid.org/0000-0003-3906-1663
Christopher Llynard Ortiz (iD) http://orcid.org/0000-0002-3114-7369
Lee-Wei Yang (iD) http://orcid.org/0000-0002-3971-6386
Qing Zhang (iD) http://orcid.org/0000-0002-6595-8995

## Ethics

All animal experiments were in compliance with National Institutes of Health guidelines and were approved by the University of Texas, Southwestern Medical Center Institutional Animal Care and Use Committee. The approved animal number protocol number is 2019-102794.

## Decision letter and Author response

Decision letter https://doi.org/10.7554/eLife.70412.sa1
Author response https://doi.org/10.7554/eLife.70412.sa2

# Additional files

## Supplementary files

• Transparent reporting form

• Supplementary file 1. Supplementary files. (a) Amplification status of ZHX2 in different breast cancer subtype. (b) The clinical information of the TNBC patient samples. (c) Top DNA-contacting residues in HD2/3/4 along with their evolutionary conservation. (d) MM/PBSA Calculations of the Free Energy (ΔH-TΔS) for Homeobox 2-4 dsDNA complexes. (e) Top C-terminal helix residues in the Homeobox 2-4 contributing the most DNA binding enthalpy. (f) Real-time PCR primers used in this study.

## Data availability

RNA-seq and ChIP-seq data are available GEO175487. All data generated or analyzed during this study are included in the manuscript and supporting files.

The following dataset was generated:

| Author(s) | Year | Dataset title | Dataset URL | Database and Identifier |
|---|---|---|---|---|
| Simon JM, Zhang Q | 2021 | ZHX2 promotes HIF1 $\alpha$ oncogenic signaling in triple-negative breast cancer | http://www.ncbi.nlm.nih.gov/geo/query/acc.cgi?acc=GSE175487 | NCBI Gene Expression Omnibus, GSE175487 |

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
