## [Editor Report]

Zinc-fingers and homeoboxes (ZHX) has been highlighted as one critical hypoxia-related factors regulator contributing to triple negative breast cancer in this work, which has enriched the upstream regulatory network of HIF-1α signaling. Moreover, identification of key residuals determining biological function of ZHX2 provides novel approaches for treating TNBC via targeting hypoxia pathway.

---

## [Decision Letter]

**Decision letter after peer review:**

Thank you for submitting your article "ZHX2 Promotes HIF1a Oncogenic Signaling in Triple-Negative Breast Cancer" for consideration by *eLife*. Your article has been reviewed by 3 peer reviewers, one of whom is a member of our Board of Reviewing Editors, and the evaluation has been overseen by Maureen Murphy as the Senior Editor. The following individual involved in review of your submission has agreed to reveal their identity: Hezhe Lv (Reviewer #3).

Essential revisions:

1) This manuscript relies heavily on in vitro assays but in vivo experimental data needs to be strengthened, and the management of figures presenting tumorigenesis requires improvement. Also, clinical significance of ZHX2 and its subcellular localization needs to be comprehensively evaluated.

2) Several associated molecules interfering with ZHX2 needs to be investigated more accurately and detailed to convey direct evidences.

3) Several downstream genes regulated by co-occupancy of ZHX2 and HIF were listed and examined for their expression level. The main pathway responsible for the oncogenic role of ZHX2 needs to be pointed out and the experimental data should be presented.

*Reviewer #1 (Recommendations for the authors):*

1. Mechanistic study was designed in a straight and narrow manner, however several conclusions were made without detailed experiment thus direct evidence lack. Also, why directly proposed that ZHX2 could be regulated by pVHL the same way as HIF in Figure 1, which seems robust, please explain this in detail.

2. ZHX2 could interfere with HIF-1α to activate downstream gene transcription, but which downstream pathway is crucial for the oncogenesis property of ZHX2? Please describe this in the manuscript in detail and add necessary experimental data.

3. Mechanistic validation lacks clinical setting, if possible you may consider to add experiment data regarding primary cells or clinical sample.

4. In Figure 4 the authors claimed that ZHX2 could co-occupy with and stabilize HIF via inhibiting the ubiquitination degradation, direct detection of the ubiquitination level should be provided.

5. In the figures manifesting western blot results of multiple groups, bar chart would better be provided to direct exhibit the expression level of the protein.

6. In Figure 3 the primary tumor mass formed by orthotropic implantation should be taken pictures and posted along statistical results, so as the lung metastases.

7. In line 273, the seven candidate positively regulated genes by HIF-1α should be annotated for their function.

*Reviewer #2 (Recommendations for the authors):*

1. Authors need to show the overexpression of ZHX2 in TNBC patient tumor tissues using IHC to confirm its subcellular localization. Authors also need to analyze the expression of pVHL in these samples through IHC. This data will establish the clinical relevance of pVHL in regulating ZHX2.

2. For all immunoprecipitation (IP) experiments, please mention how much input was used. As the size of pVHL lies in between 24-30 kDa, and the light chain band also corresponds to approximately 25 kDa, in Figure 1G, please show the full western blot with heavy and light chain bands of IgG.

3. Authors have used only one cell line for in-vivo studies and are required to use at least two different breast cancer cell lines for experimentation to justify the significance of outcome.

4. Authors should provide bioluminescence images of mice day after injection and on the day of tumor harvesting, which is indicative of equal number of cell injections and helps in visualizing the difference in tumor growth and metastasis.

5. Did the authors analyze the slower rate of tumor growth in ZHX2-knockdown group is due to apoptosis or senescence using IHC for cleaved caspase 3 or p21 antibody?

6. Authors should provide representative photographs of tumors harvested from different animal groups. In addition, please provide the representative photographs or H and E images of lungs to compare the number and size of metastatic nodules between different groups.

7. Authors are required to evaluate the expression of proliferative (Ki-67) angiogenic markers such as CD31 or CD34 in the xenograft tumors.

8. As the manuscript falls short of making a compelling case for the mechanism described to be relevant in the clinical setting, authors should evaluate the clinical significance of ZHX2 on TNBC growth and metastasis using TNBC PDX models.

9. Authors should perform the luciferase activity in a dose dependent manner by increasing the concentration of ZHX2-expressing plasmid to assess the HIF promoter activity. Authors should include one more group in the assay: ZHX2res alone.

10. For the ease of readers, authors should provide a schematic diagram highlighting the novel findings.

*Reviewer #3 (Recommendations for the authors):*

1. Authors have demonstrated that ZHX2 promoted NF-κB activation and acted as an oncogenic driver in clear cell renal cell carcinoma. Is there any different function of ZHX2 in TNBC and clear cell renal cell carcinoma? What is the relationship between NF-κB pathway and ZHX2 in TNBC?

2. The authors presented ZHX2 and c-Myc share co-amplification in many cancer types. Is there any relationship between ZHX2 and c-Myc?

3. Throughout the manuscript that there are many syntactic errors and the labeling in some Figures is unclear (eg. Figure 4). For example, "Figure 4-Figure supplement 4C", "shZHX2_DN or shZHX2_DE" are hard to understand what they mean, and the Figure legend for Figure 4K is missing.

4. In Figure 1F, the picture of Ponceau red staining is not correct. The last line is marker.

5. For some IP staining, like Figure 1G, Figure 4E-4G, the whole cell lysates control is missing.

6. In Figure 3, the number of mice with metastasis tumor is missing, and what's the difference between Figure 3F and Fig3K?

7. In Figure 6G, why does ZHX2 mutant lead to decreased HIFa target genes without affecting HIFa binding and protein level? The binding ability between HIFa and other ZHX2 mutants should be tested.

[Editors' note: further revisions were suggested prior to acceptance, as described below.]

Thank you for submitting your article "ZHX2 Promotes HIF1a Oncogenic Signaling in Triple-Negative Breast Cancer" for consideration by *eLife*. Your article has been reviewed by 3 peer reviewers, one of whom is a member of our Board of Reviewing Editors, and the evaluation has been overseen by Wafik El-Deiry as the Senior Editor. The reviewers have opted to remain anonymous.

Essential revisions:

1) Data and images manifesting the role of ZHX2 on triple negative breast cancer metastasis need to be revised in Figure 3;

2) Clinical information of the included TNBC patients should better be summarized into one table and presented in the supplement.

*Reviewer #1 (Recommendations for the authors):*

The authors have made corresponding adjustments according to reviewers' comments and improved the quality much, however, there still exist several main concerns regarding the figures.

1. Expression level of ZHX2 in some of the 10 included TNBC patients is different to those provided in the initial version, and clinical information of the patients should better be provided in supplementary material;

2. Original image of western blot assay should be addressed without too much manipulation;

3. In vivo and ex vivo data exhibiting the role of ZHX2 exert on MDA-MB-231 metastasis capacity should be re-organized and more rigor;

4. Some of the replicative data in the main figure should be moved to supplementary part to achieve more clear manifestation of the results.

*Reviewer #2 (Recommendations for the authors):*

Authors have considerably improved the manuscript. However, a few things need to be clarified.

1. In supplement figure 1D, authors have knocked out VHL using different sgRNAs but even in ctrl cells there is no expression of VHL. The image is too much manipulated for brightness. Authors are requested not to modify the WB images too much (background is too white) because it makes it hard to interpret the data.

2. In-vivo metastasis results are not clear, just ex-vivo BLI analysis is not sufficient to justify the metastatic role of ZHX2. It is well established that MDA-MB-231 cells highly metastasize to lungs (recent paper: PMID: 34465361). Authors were requested to provide H and E of whole lungs or a major lobe of lung to discern the metastatic effect. This way, authors can even highlight micro metastasis in inset of the figure.

*Reviewer #3 (Recommendations for the authors):*

The authors have done a good job in addressing my prior concerns. I am satisfied with the responses to my criticisms/comments except one major question.

Fig1F. ZHX2 expression pattern is opposite in several patients compared with previous version. Authors need to provide patients' information in supplemental tables.

---

## [Author Response]

Essential revisions:1) This manuscript relies heavily on in vitro assays but in vivo experimental data needs to be strengthened, and the management of figures presenting tumorigenesis requires improvement. Also, clinical significance of ZHX2 and its subcellular localization needs to be comprehensively evaluated.

We appreciate the constructive comments from the editor and reviewers.

We have used a second TNBC cell line HCC70 which expresses the highest ZHX2 protein level according to our western blot of cell panel from Figure 1E. We generated the inducible Teton shRNA HCC70-luciferase cell lines for control, ZHX2 sh43 and sh45. We have verified the knockdown efficiency by western blot as well as phenotype in these cell lines with or without doxycycline treatment, as showed in the new Figure 3B. We have injected 1E6 cells orthotopically into NSG mice and 10 mice for each group. Upon tumors were formed (50~100mm^3^) all mice were fed with doxycycline diet. We monitor tumor growth over time for 5 weeks after cell inoculation, tumors were dissected at the same time. Consistent with the finding from MDA-MB-231 cell line (revised Figure 3A, G, K and L), we observed significant decrease of tumorigenesis (reflected by tumor growth and harvest tumor weight) in the two ZHX2 shRNA knockdown HCC70 cells compared with the control shRNA (new Figure 3H, M and N). In addition to monitor tumor growth by caliper, as requested by reviewer, we also performed live bioluminescence imaging over time and provide representative image of initial and end of the experiment (new Figure 3I), we found that the relative bioluminescence signal intensity is consistent with the tumor volume growth (new Figure 3J).

We also provide the image for the harvest tumor from our previous mice experiment using MDA-MB-231 xenograft (new Figure 3K, L) as requested by the reviewer.

Figure. 3

To improve the clinical significance of ZHX2, we obtained 20 pairs of fresh cut normal/tumor TNBC patient slides from the tissue management core facility from UTSW and performed IHC staining with these slides, we found that an overall increased ZHX2 staining in the TNBC tumors, especially in the nuclear staining by quantification (new Figure 1G, H), suggesting overall and the nuclear located ZHX2 is upregulated in tumors from TNBC patients. In addition, we have made new samples and ran the western blot from 10 pairs of normal/tumor TNBC patient tissues, we found that 8 out of 10 samples showed increased ZHX2 expression (revised Figure 1F).

2) Several associated molecules interfering with ZHX2 needs to be investigated more accurately and detailed to convey direct evidences.

Although ZHX2 and c-Myc have similar amplification and expression manner in breast cancer (Figure A, B and E). Our data showed that knockdown ZHX2 did not affect protein level of c-Myc, suggesting the oncogenic function of ZHX2 is independent of c-Myc (revised Figure 3A, B).

We previous find that ZHX2 promoted nuclear factor κB (NF-κB) activation by directly binding with p65 and promoting its nuclear translocation in clear cell renal cell carcinoma (ccRCC) (Zhang et al., 2018). We also investigated the relationship between ZHX2 and p65 in breast cancer. We first examined the interaction of ZHX2 and p65 in breast cancer by endogenous IP with ZHX2 or p65 antibody in two TNBC cells (MDA-MB-231 and 468) (new Figure 4—figure supplement 1A, B), we did not find consistent and robust binding between ZHX2 and p65 as we observed in ccRCC. In addition, knockdown ZHX2 also did not change the gross p65 protein level (new Figure 4—figure supplement 1C), or the subcellular protein level of p65 either in cytosol or nucleus (new Figure 4—figure supplement 1D, E). Furthermore, we did not find the NF-κB pathway enriched through our RNA-seq analysis (Figure 4—figure supplement 2A). Altogether, these data suggest ZHX2 may not affect p65 translocation in TNBC. However, these preliminary data could not fully exclude the possibility that ZHX2 regulates the NF-κB signaling pathway via other factor components, which needs further investigation.

In our current study, we found that knockdown of ZHX2 affected the HIF signaling more consistently than other pathways. This gave us the rationale to examine the relationship between ZHX2 and HIF signaling. We found that ZHX2 directly interacts with and promotes stability of HIF1α, but not HIF2α (revised Figure 4E-N). On the contrary, HIF1α knockdown does not affect ZHX2 protein level (revised Figure 4—figure supplement 2H). We also identified 7 potential direct downstream genes that could be co-occupied and regulated by ZHX2 and HIF1α (Figure. 5). In this revision, we examined the direct downstream targets that responsible for the oncogenic role of ZHX2 (See response for major point 3).

3) Several downstream genes regulated by co-occupancy of ZHX2 and HIF were listed and examined for their expression level. The main pathway responsible for the oncogenic role of ZHX2 needs to be pointed out and the experimental data should be presented.

We appreciate the constructive comments from the editor and reviewers. We have identified 7 direct downstream targets of ZHX2 and HIF1α (Figure 5A-D). To determine which gene is essential for maintaining the oncogenic function of ZHX2, we performed rescue experiments by overexpressing these downstream target genes. We got six plenti6.3 based expression constructs (PTGES3L, KDM3A, WSB1, AP2B1, OXSR1 and COX20) from DNASU (https://dnasu.org/DNASU/Home.do) except RUNDC1, and verified these plasmids by sequencing before use. We decide to perform rescue experiment for these 6 genes. Upon generating stable blasticidin selected cell lines for each gene in MDA-MB-231, we subsequently knocked down ZHX2 by sh45 in each stable cell lines and selected by puromycin for 4 days followed by performing 2-D colony formation and 3-D soft agar growth. We eventually found that 5 genes expressed with expected size except OXSR1 in these selected stable cell lines (new Figure 5E). We performed two independent 2-D colony formation assays and found that at least 4 genes (AP2B1, COX20, KDM3A, and PTGES3L) could partially rescue the growth by ZHX2 depletion (new Figure 5F, G). As expected, the cell line failed to express OXSR1 could not rescue the phenotype. The partially rescue phenotype suggested that these downstream targets contribute to the oncogenic role of ZHX2 in an accumulative fashion.

Figure 5

Reviewer #1 (Recommendations for the authors):1. Mechanistic study was designed in a straight and narrow manner, however several conclusions were made without detailed experiment thus direct evidence lack. Also, why directly proposed that ZHX2 could be regulated by pVHL the same way as HIF in Figure 1, which seems robust, please explain this in detail.

We thanks the comments from the reviewer. We first demonstrated that ZHX2 is subjected to VHL regulation in renal cancer (Zhang et al., 2018). We want to establish the boarder implication that VHL can regulate ZHX2 protein stability in different cancer contexts, including renal cancer and breast cancer. HIF is a well know substrate of VHL, and HIF1α plays important role in breast cancer tumorigenesis (Briggs et al., 2016; Chen et al., 2014). We want to emphasis that ZHX2 and HIF1α both are important targets of VHL, indicating a similar upstream regulatory mechanism in TNBC. More importantly, we found that ZHX2 directly bound and regulated HIF1α in this study, suggesting ZHX2 is oncogenic important for TNBC.

2. ZHX2 could interfere with HIF-1α to activate downstream gene transcription, but which downstream pathway is crucial for the oncogenesis property of ZHX2? Please describe this in the manuscript in detail and add necessary experimental data.

We appreciate the constructive comments from the reviewer. We have identified 7 direct downstream targets of ZHX2 and HIF1α (Figure 5A-D). To determine which gene is essential for maintaining the oncogenic function of ZHX2, we performed rescue experiments by overexpressing these downstream target genes, we got 6 plenti6.3 based expression constructs (PTGES3L, KDM3A, WSB1, AP2B1, OXSR1 and COX20) from DNASU (https://dnasu.org/DNASU/Home.do) except RUNDC1, we verified these plasmids by sequencing before use. We decide to perform rescue experiment for these 6 genes, upon generating stable blasticidin selected cell lines for each gene in MDA-MB-231, we subsequently knockdown ZHX2 by sh45 in each stable cell lines and selected by puromycin for 4 days followed by performing 2-D colony formation and 3-D soft agar growth. We eventually found that 5 genes expressed with expected size except OXSR1 in these selected stable cell lines (new Figure 5E). We performed 2-D colony formation assay and showed that at least 4 genes (AP2B1, COX20, KDM3A, and PTGES3L) could partially rescue the growth by ZHX2 depletion (new Figure 5F and G). As expected, the cell line failed to express OXSR1 could not rescue the phenotype. The partially rescue phenotype suggested that these downstream targets contribute to the oncogenic role of ZHX2 in an accumulative fashion.

3. Mechanistic validation lacks clinical setting, if possible you may consider to add experiment data regarding primary cells or clinical sample.

To improve the clinical significance of ZHX2, we got 20 pairs of fresh cut normal/tumor TNBC patient slides from the tissue management core facility from UTSW and performed IHC staining with these slides, we found that an overall increased ZHX2 staining in the TNBC tumors, especially in the nuclear staining by quantification (see new Figure 1G and H), suggesting ZHX2 is upregulated in tumors from TNBC patients. In addition, we have made new samples and ran the western blot from 10 pairs of normal/tumor TNBC patient tissues, we found that 8 out of 10 samples showed increased ZHX2 expression (see revised Figure 1F).

4. In Figure 4 the authors claimed that ZHX2 could co-occupy with and stabilize HIF via inhibiting the ubiquitination degradation, direct detection of the ubiquitination level should be provided.

We have performed a ubiquitination assay by IP pull-down endogenous HIF1α from the cell lysate from empty vector (EV) or ZHX2 overexpression MDA-MB-231 cells, we detected significant less ubiquitination of HIF1α in ZHX2 overexpression cells than the EV control cells (new Figure 4O), suggesting ZHX2 could protect HIF1α from the proteasome degradation.

5. In the figures manifesting western blot results of multiple groups, bar chart would better be provided to direct exhibit the expression level of the protein.

We thanks reviewer for this good suggestion. We have quantified western blot bands where quantification is needed by using ImageJ, we have added the quantification values blow some important blots instead of bar chart due to limited figure space. These figures including revised or new Figures 1J-L, Figure 1—figure supplement 1D and E, Figure 4I-L, 4N,4O.

6. In Figure 3 the primary tumor mass formed by orthotropic implantation should be taken pictures and posted along statistical results, so as the lung metastases.

We have provided images of harvested tumors from the two animal experiments using MDA-MB-231 and HCC70 xenografts, please see revised Figure 3K and M. We have provided tumor weight of these tumors which showed statistical significance between the shRNA control and ZHX2 KD groups (Figure 3L and N). For the lung metastases, we performed bioluminescence imaging of the lungs from MDA-MB-231 since it is a metastatic cell line than HCC70 (Figure 3P and Q), however, we didn’t observe notable tumor on the surface of lung, possibly due to a short term of tumor implantation (4 weeks), therefore, we didn’t take bright light image of lung. We also didn’t observe detectable lung bioluminescence signal in the HCC70 xenograft at the end point.

7. In line 273, the seven candidate positively regulated genes by HIF-1α should be annotated for their function.

We have annotated the information of these 7 genes in the text.

KDM3A (Lysine Demethylase 3A) is a zinc finger protein that contains a jumonji domain and had identified as the histone H3 lysine 9 mono- and di-methyl demethylase enzyme(Wang et al., 2013). KDM3A is overexpressed in breast cancer tissues (Yoo et al., 2020). WSB1 was the E3 ubiquitin ligase and has an intriguing role in regulating immune responsiveness, glycolysis, hypoxia, development or growth of cancer (Haque et al., 2016). Dysregulation of WSB1 expression has been documented in diverse cancer cell types, including breast cancer, neuroblastoma, hepatocellular carcinoma, pancreatic, and osteosarcoma (Haque et al., 2016). OXSR1 belongs to the Ser/Thr protein kinase family of proteins and regulates downstream kinases in response to environmental stress and may play a role in regulating the actin cytoskeleton (Li et al., 2020). COX20 (also known as FAM36A; MIM#614698) is a mitochondrial inner membrane protein, whose known function is to chaperone COX2, a subunit of cytochrome c oxidase in the yeast’s mitochondrial matrix (Hell et al., 2000; Otero et al., 2019). It contains two transmembrane helices and localizes to the mitochondrial membrane. Mutations in this gene can cause mitochondrial complex IV deficiency, which results in ataxia and muscle hypotonia(Otero et al., 2019).(Keerthiraju et al., 2019) AP2B1 is a component of the adaptor protein complex 2 (AP-2) (Lau and Chou, 2008). This gene encodes a member of the AP-2 family of transcription factors. AP-2 proteins form homo- or hetero-dimers with other AP-2 family members and bind specific DNA sequences. They are thought to stimulate cell proliferation and suppress terminal differentiation of specific cell types during embryonic development. RUNDC1 contains a RUN (RPIP8, UNC-14 and NESCA) domain and a coiled coil domain. The encoded protein may negatively regulate p53 transcriptional activity (Llanos et al., 2006). PTGES3L (Prostaglandin E Synthase 3-Like) is predominantly expressed in skeletal muscle. Diseases associated with PTGES3L include arthrogryposis, distal, type 2A and spondylocarpotarsal synostosis syndrome. Limited research has explored the function of PTGES3L in tumors.Reviewer #2 (Recommendations for the authors):1. Authors need to show the overexpression of ZHX2 in TNBC patient tumor tissues using IHC to confirm its subcellular localization. Authors also need to analyze the expression of pVHL in these samples through IHC. This data will establish the clinical relevance of pVHL in regulating ZHX2.

To improve the clinical significance of ZHX2, we got 20 pairs of fresh cut normal/tumor TNBC patient slides from the tissue management core facility from UTSW and performed IHC staining with these slides, we found that an overall increased ZHX2 staining in the TNBC tumors, especially in the nuclear staining by quantification (see new Figure 1G and H), suggesting overall and the nuclear located ZHX2 is upregulated in tumors from TNBC patients. In addition, we have made new samples and ran the western blot from 10 pairs of normal/tumor TNBC patient tissues, we found that 8 out of 10 samples showed increased ZHX2 expression (see revised Figure 1F).

To will establish the clinical relevance of pVHL with ZHX2, we obtained two commercial TNBC tissue microarrays (TMA) from US Biomax (BR1301), which contains 130 samples hopefully to gain sufficient clinical correlation. We IHC stained ZHX2 or pVHL for each TMA respectively (new Figure 1-figure supplement 1G). We quantified the staining intensity and gave IHC different staining level according to the staining intensity to each sample. From the quantitative analysis, we found an overall negative correlation between pVHL and ZHX2 in these TNBC samples (new Figure 1M and N).

2. For all immunoprecipitation (IP) experiments, please mention how much input was used. As the size of pVHL lies in between 24-30 kDa, and the light chain band also corresponds to approximately 25 kDa, in Figure 1G, please show the full western blot with heavy and light chain bands of IgG.

We used 2% input for original IP, in this revised IP experiment, we increased the input loading to 5% to increase the visibility of some input bands. We have provided the information in the revised figure legends.

In Figure 1G and Figure 1—figure supplement 1D, we overexpressed HA-ZHX2 or EV in MDA-MB-231 cells and MDA-MB-468 cells, then we did IP by anti-HA magnetic beads (Thermo Scientific, 88836). Anti-HA magnetic beads hydroxyl magnetic beads covalently coupled with mouse IgG3a monoclonal. Rabbit anti-ZHX2 antibody (Genetex, 112232) and rabbit anti-VHL (Cell Signaling, 68547) were used for the Western Blotting. Goat anti-rabbit secondary antibody were used as the second antibody, and cannot recognize the mouse IgG, so the heavy and light chain band of mouse IgG cannot be displayed in the WB figure. Hence, the light chain would not affect the band of pVHL.

3. Authors have used only one cell line for in-vivo studies and are required to use at least two different breast cancer cell lines for experimentation to justify the significance of outcome.

We appreciate the constructive comments from the editor and reviewers.

In this revision, We have used a second TNBC cell line HCC70 which expresses the highest ZHX2 protein level according to our western blot of cell panel from Figure 1E. We generated the inducible Teton shRNA HCC70-luciferase cell lines for control, ZHX2 sh43 and sh45. We have verified the knockdown efficiency by western blot as well as phenotype in these cell lines with or without doxycycline treatment, as showed in the new Figure 3B. We have injected 1E6 cells orthotopically into NSG mice and 10 mice for each group. Upon tumors were formed (50~100mm^3^) all mice were fed with doxycycline diet. We monitor tumor growth over time for 5 weeks after cell inoculation, tumors were dissected at the same time. Consistent with the finding from MDA-MB-231 cell line (revised Figure 3A, G, K and L), we observed significant decrease of tumorigenesis (reflected by tumor growth and harvest tumor weight) in the two ZHX2 shRNA knockdown HCC70 cells compared with the control shRNA (new Figure 3H, M and N).

4. Authors should provide bioluminescence images of mice day after injection and on the day of tumor harvesting, which is indicative of equal number of cell injections and helps in visualizing the difference in tumor growth and metastasis.

In addition to monitor tumor growth by caliper, as requested by reviewer, we also performed live bioluminescence imaging over time and provide representative image of initial and end of the experiment (new Figure 3I), we found that the relative bioluminescence signal intensity is consistent with the tumor volume growth (new Figure 3J).

5. Did the authors analyze the slower rate of tumor growth in ZHX2-knockdown group is due to apoptosis or senescence using IHC for cleaved caspase 3 or p21 antibody?

We thank the comments from reviewer. We have chosen four representative tumors from each group from the harvested MDA-MB-231 xenograft tumors, performed histology and IHC staining. We performed H&E and stained ZHX2, Ki-67 (proliferative marker), Endomucin (angiogenic marker), and cleaved-caspase3 (apoptosis) for these slides. As suggested by the staining, we found that ZHX2 depletion mainly affects proliferation for the tumor cells reflected by marked decrease of Ki-67 in ZHX2 KD tumor, rather than angiogenesis or apoptosis (new Figure 3L, M).

6. Authors should provide representative photographs of tumors harvested from different animal groups. In addition, please provide the representative photographs or H and E images of lungs to compare the number and size of metastatic nodules between different groups.

We have provided images of harvested tumors from the two animal experiments using MDA-MB-231 and HCC70 xenografts, please see revised Figure 3K and M. We have provided tumor weight of these tumors which showed statistical significance between the shRNA control and ZHX2 KD groups (Figure 3L, N). For the lung metastases, we performed bioluminescence imaging of the lungs from MDA-MB-231 since it is a metastatic cell line than HCC70 (Figure 3P, Q), however, we didn’t observe notable tumor on the surface of lung, possibly due to a short term of tumor implantation (4 weeks), therefore, we didn’t take bright light image of lung. We also didn’t observe detectable lung bioluminescence signal in the HCC70 xenograft at the end point.

7. Authors are required to evaluate the expression of proliferative (Ki-67) angiogenic markers such as CD31 or CD34 in the xenograft tumors.

We thank the comments from reviewer. We have stained Ki-67 as a proliferative maker and Endomucin as an angiogenic marker. These results belong to a same batch of staining. Please refer our response to your comment 5.

8. As the manuscript falls short of making a compelling case for the mechanism described to be relevant in the clinical setting, authors should evaluate the clinical significance of ZHX2 on TNBC growth and metastasis using TNBC PDX models.

We thanks the reviewer for the suggestions using patient-derived xenograft (PDX) models. It is technically difficult to manipulate ZHX2 in PDX tumors, plus considering the PDX expanding will take plenty of time, we will not address this at the current stage, which will be added to our future directions.

9. Authors should perform the luciferase activity in a dose dependent manner by increasing the concentration of ZHX2-expressing plasmid to assess the HIF promoter activity. Authors should include one more group in the assay: ZHX2res alone.

We thank the reviewer for this suggestion. Sub-confluent HEK293T cells (200,000 cells/24‐well plate) were transiently transfected with different amount of HA-ZHX2 (100 ng, 200 ng, and 400 ng), 100 ng HRE-luci reporter and 30 ng pCMV-*Renilla*. Forty‐eight hours after transfection, luciferase assays were performed by Dual-Luciferase Reporter Assay System (Promega, E1960). We found that increasing the concentration of ZHX2-expressing plasmid could increasing the HIF reporter activity.

10. For the ease of readers, authors should provide a schematic diagram highlighting the novel findings.

We drew a schematic diagram to summary the major findings of this study. We have incorporated this schematic to the new Figure 6J.

Reviewer #3 (Recommendations for the authors):1. Authors have demonstrated that ZHX2 promoted NF-κB activation and acted as an oncogenic driver in clear cell renal cell carcinoma. Is there any different function of ZHX2 in TNBC and clear cell renal cell carcinoma? What is the relationship between NF-κB pathway and ZHX2 in TNBC?

Thanks for the good suggestion. We previous find that ZHX2 promoted nuclear factor κB (NF-κB) activation by directly binding with p65 and promoting its nuclear translocation in clear cell renal cell carcinoma (ccRCC) (Zhang et al., 2018). As suggested by the reviewer, we also investigated the relationship between ZHX2 and p65 in breast cancer. We first examined the interaction of ZHX2 and p65 in breast cancer by endogenous IP with ZHX2 or p65 antibody in two TNBC cells (MDA-MB-231 and 468) (new Figure 4—figure supplement 1A and B), we did not find consistent and robust binding between ZHX2 and p65 as we observed in ccRCC. In addition, knockdown ZHX2 also did not change the gross protein level of p65 (new Figure 4—figure supplement 1C), or the subcellular protein level of p65 either in cytosol or nucleus (new Figure 4—figure supplement 1D and E). Furthermore, in our ZHX2 CHIP seq, ZHX2 did not bind with the promoter or enhancer of p65 genes. We also did not find NF-κB pathway has been enriched through our RNA-seq analysis (revised Figure 4—figure supplement 2A). Altogether, these data suggest ZHX2 may not affect p65 translocation in TNBC. However, these preliminary data could not fully exclude the possibility that ZHX2 regulates the NF-κB signaling pathway via other factor components, which needs further investigation.

In our current study, we found that knockdown of ZHX2 affected the HIF signaling most than other pathways. This gave us the rationale to examine the relationship between ZHX2 and HIF. We found that ZHX2 directly interacts with and promotes stability of HIF1α, but not HIF2α (Figure 4E – M). On the contrary, HIF1α knockdown does not affect ZHX2 protein level (revised Figure 4—figure supplement 2F). We also identified 7 potential direct downstream genes that could be co-occupied and regulated by ZHX2 and HIF1α (Figure 5). In this revision, we will examine the direct downstream targets that responsible for the oncogenic role of ZHX2 (See response for major point 3).

2. The authors presented ZHX2 and c-Myc share co-amplification in many cancer types. Is there any relationship between ZHX2 and c-Myc?

We thanks for the reviewer for the important comment. *ZHX2* is located on 8q24, where c-*Myc* resides. Analysis of copy number across different cancer types from The Cancer Genome Atlas (TCGA) showed that, *ZHX2* and c-*Myc* share co-amplification in most cancer types observed (Figure 1B). In our ZHX2 RNA-seq, ZHX2 KD did not influence the mRNA level of Myc. In the ChIP-seq, ZHX2 did not bind with the promoter or enhancer of Myc genes. In addition, ZHX2 KD did not lead to c-Myc protein expression difference (Figure 3A, H). Taken together, these evidences indicate no regulatory relationship between ZHX2 and c-Myc.

3. Throughout the manuscript that there are many syntactic errors and the labeling in some Figures is unclear (eg. Figure 4). For example, "Figure 4-Figure supplement 4C", "shZHX2_DN or shZHX2_DE" are hard to understand what they means, and the Figure legend for Figure 4K is missing.

Sorry for the mistakes. We have corrected all these mistakes.

In our original labelling in Figure 4A, “ZHX2 sh43 vs shCtrl_DOWN DEGs” means the downregulated differentially expressed genes (DEGs) when cells were knockdown by ZHX2 sh43 compared to control. “ZHX2 sh43 vs shCtrl_UP DEGs” means the upregulated differentially expressed genes when cells were knockdown by ZHX2 sh43 compared to control. In Figure 4C, DE means differentially expression, DN means down. “HIF DKO VS WT DE” means the DEGs when cells were double knockdown by HIF compared to the wild type. “HIF DKO VS WT DE” means the downregulated genes when cells were double knockdown by HIF compared to the wild type.

To make the labels simple for reader, we have redrawn these figures. Please see below revised Figure 4A, B, D and E. We have added the abbreviations in the figure legends.

We have added the figure legend for Figure 4K: Immunoprecipitations of cell lysates from MDA-MB-231 cells infected with lentivirus encoding EV or FLAG-ZHX2.

4. In Figure 1F, the picture of Ponceau red staining is not correct. The last line is marker.

We have made new samples and ran the western blot from 10 pairs of normal/tumor TNBC patient tissues, we found that 8 out of 10 samples showed increased ZHX2 expression (see revised Figure 1F).

5. For some IP staining, like Figure 1G, Figure 4E-4G, the whole cell lysates control is missing.

Thanks for point these out. For the original Figure 1G, 4E, 4F and 4G, we have redone the IP experiment by increasing the input to 5%. Please see the new Figure 1I, 4F, G and H and Figure 4—figure supplement 2G.

6. In Figure 3, the number of mice with metastasis tumor is missing, and what's the difference between Figure 3F and Fig3K?

Thanks for the comments. We used bioluminescence image to indicate the metastatic burden in the lung. However, from the bioluminescence signal, it hard to precisely exclude mice without any lung metastasis, because the signal could be too weak to determine. From the quantification value with all > 0 suggesting all mice may have lung metastasis, with the possibility of background signal. There is significant signal decrease upon ZHX2 knockdown (revised Figure 3P and Q), suggesting ZHX2 promotes lung metastasis. We didn’t count the metastasis site in lung, due to short term cell line implantation during the whole animal experiment (4 weeks), we didn’t observe obvious metastatic tumor on the surface of lung.

Data from the original Figure 3F and Figure 3K represents two different sets of experiments, in Figure 3F, we compared the tumor growth between two ZHX2 shRNAs with the control shRNA. In Figure 3K, we injected the same Teton sh45 MDA-MB-231 cell line in 10 mice, then we randomized the 10 mice into two groups (5/group) with or without the doxycycline diet treatment. We realize that the second set of experiment is redundant and caused some confusion for the reader. Given we have added new animal experiment with another cell line. We thus decided to remove the data from the original Figure 3K-M.

7. In Figure 6G, why does ZHX2 mutant lead to decreased HIFa target genes without affecting HIFa binding and protein level? The binding ability between HIFa and other ZHX2 mutants should be tested.

Thanks for the good comments. We speculate that ZHX2 binds to these genes independently from HIF1a, which means ZHX2 or HIF1a could bind to these genes in an independent manner. ZHX2 could interact and stabilize HIF1a, but ZHX2 is not necessary depended on the HIF1a to execute its oncogenic function. On the other hand, these mutant sites were computed for interaction site of ZHX2 with DNA, theoretically their mutation will not affect the binding with HIF1a, which likely relies on other amino acid residues. As a test of the speculation, we examined the two most important residues (681, 674) which control the oncogenic role of ZHX2 and found no binding loss of these two mutants with HIF1a (Figure 6G). We feel that examine the binding between HIF1a and other ZHX2 mutants will not add further mechanistic insight to this study. Therefore, we will not examine more interaction between HIF1a and other ZHX2 mutants. However, these speculations need further investigation, we will address these important questions in our future study.

References:

Briggs KJ, Koivunen P, Cao S, Backus KM, Olenchock BA, Patel H, Zhang Q, Signoretti S, Gerfen GJ, Richardson AL, et al. 2016. Paracrine Induction of HIF by Glutamate in Breast Cancer: EglN1 Senses Cysteine. Cell 166: 126-139. https://doi.org/10.1016/j.cell.2016.05.042.

Chen X, Iliopoulos D, Zhang Q, Tang Q, Greenblatt MB, Hatziapostolou M, Lim E, Tam WL, Ni M, Chen Y, et al. 2014. XBP1 promotes triple-negative breast cancer by controlling the HIF1alpha pathway. Nature 508: 103-107. https://doi.org/10.1038/nature13119.

Haque M, Kendal JK, MacIsaac RM, and Demetrick DJ. 2016. WSB1: from homeostasis to hypoxia. Journal of biomedical science 23: 61. https://doi.org/10.1186/s12929-016-0270-3.

Hell K, Tzagoloff A, Neupert W, and Stuart RA. 2000. Identification of Cox20p, a novel protein involved in the maturation and assembly of cytochrome oxidase subunit 2. The Journal of biological chemistry 275: 4571-4578. https://doi.org/10.1074/jbc.275.7.4571.

Keerthiraju E, Du C, Tucker G, and Greetham D. 2019. A Role for COX20 in Tolerance to Oxidative Stress and Programmed Cell Death in *Saccharomyces cerevisiae*. Microorganisms 7. https://doi.org/10.3390/microorganisms7110575.

Lau AW, and Chou MM. 2008. The adaptor complex AP-2 regulates post-endocytic trafficking through the non-clathrin Arf6-dependent endocytic pathway. Journal of cell science 121: 4008-4017. https://doi.org/10.1242/jcs.033522.

Lee CK, Jeong SH, Jang C, Bae H, Kim YH, Park I, Kim SK, and Koh GY. 2019. Tumor metastasis to lymph nodes requires YAP-dependent metabolic adaptation. Science 363: 644-649. https://doi.org/10.1126/science.aav0173.

Li Y, Qin J, Wu J, Dai X, and Xu J. 2020. High expression of OSR1 as a predictive biomarker for poor prognosis and lymph node metastasis in breast cancer. Breast cancer research and treatment 182: 35-46. https://doi.org/10.1007/s10549-020-05671-w.

Llanos S, Efeyan A, Monsech J, Dominguez O, and Serrano M. 2006. A high-throughput loss-of-function screening identifies novel p53 regulators. Cell cycle (Georgetown, Tex) 5: 1880-1885. https://doi.org/10.4161/cc.5.16.3140.

Otero MG, Tiongson E, Diaz F, Haude K, Panzer K, Collier A, Kim J, Adams D, Tifft CJ, Cui H, et al. 2019. Novel pathogenic COX20 variants causing dysarthria, ataxia, and sensory neuropathy. Annals of clinical and translational neurology 6: 154-160. https://doi.org/10.1002/acn3.661.

Subramanian A, Tamayo P, Mootha VK, Mukherjee S, Ebert BL, Gillette MA, Paulovich A, Pomeroy SL, Golub TR, Lander ES, et al. 2005. Gene set enrichment analysis: a knowledge-based approach for interpreting genome-wide expression profiles. Proc Natl Acad Sci U S A 102: 15545-15550. https://doi.org/10.1073/pnas.0506580102.

Wang L, Chang J, Varghese D, Dellinger M, Kumar S, Best AM, Ruiz J, Bruick R, Peña-Llopis S, Xu J, et al. 2013. A small molecule modulates Jumonji histone demethylase activity and selectively inhibits cancer growth. Nature communications 4: 2035. https://doi.org/10.1038/ncomms3035.

Yoo J, Jeon YH, Cho HY, Lee SW, Kim GW, Lee DH, and Kwon SH. 2020. Advances in Histone Demethylase KDM3A as a Cancer Therapeutic Target. Cancers 12. https://doi.org/10.3390/cancers12051098.

Zhang J, Wu T, Simon J, Takada M, Saito R, Fan C, Liu XD, Jonasch E, Xie L, Chen X, et al. 2018. VHL substrate transcription factor ZHX2 as an oncogenic driver in clear cell renal cell carcinoma. Science (New York, NY) 361: 290-295. https://doi.org/10.1126/science.aap8411.

[Editors' note: further revisions were suggested prior to acceptance, as described below.]

Essential revisions:1) Data and images manifesting the role of ZHX2 on triple negative breast cancer metastasis need to be revised in Figure 3;

We appreciate the constructive comments from the editor and reviewers. We have reorganized the Figure 3, we also moved some of the repetitive data to the supplementary files. Please see updated Figure 3 and supplemental figures in the revised manuscript.

2) Clinical information of the included TNBC patients should better be summarized into one table and presented in the supplement.

We appreciate the constructive comments from the editor and reviewers. The Clinical information of the included TNBC patients have been provided in the table Supplementary File 1b.

Reviewer #1 (Recommendations for the authors):The authors have made corresponding adjustments according to reviewers' comments and improved the quality much, however, there still exist several main concerns regarding the figures.

1. Expression level of ZHX2 in some of the 10 included TNBC patients is different to those provided in the initial version, and clinical information of the patients should better be provided in supplementary material;

We thank the comments from reviewer. In our initial western blot of the TNBC patient samples, we accidently loaded the marker together with one of the patient samples as shown in the Ponceau red staining. In the revision, we tried to rerun the western blot samples however the Ponceau red staining suggested most of the proteins were degraded during the long-term storage. Therefore, we freshly extracted new lysates from the same 10 pairs of patient frozen tissue samples and ran western blotting but in a different order of the sample organization from our initial version. Therefore, the expression pattern looks different from the initial version. We are sorry for making this confusion to the reviewers but we want to note that the overall expression of ZHX2 in these 10 pairs TNBC samples from biological replicate experiments is quite similar.

The clinical information of the included TNBC patients have been provided in the table Supplementary File 1b.

2. Original image of western blot assay should be addressed without too much manipulation;

We thanks for the reviewer for the critical suggestion. We have adjusted the western blot images throughout the figures to make sure they are consistent with the original format. Please check our updated figures. In addition, we also have attached all of original source data for western blots to ensure the consistency.

3. In vivo and ex vivo data exhibiting the role of ZHX2 exert on MDA-MB-231 metastasis capacity should be re-organized and more rigor;

We thank the comments from reviewer. We have re-organized the original Figure 3 to make the data clearer to the reader.

4. Some of the replicative data in the main figure should be moved to supplementary part to achieve more clear manifestation of the results.

We thank the excellent suggestion from reviewer. We have moved some data from the main figures to the supplemental figures. These include the original Figure 3C-F, M-N; Figure 4G. Please check our new Figure 3 and 4, as well as the corresponding supplemental figures.

Reviewer #2 (Recommendations for the authors):Authors have considerably improved the manuscript. However, a few things need to be clarified.1. In supplement figure 1D, authors have knocked out VHL using different sgRNAs but even in ctrl cells there is no expression of VHL. The image is too much manipulated for brightness. Authors are requested not to modify the WB images too much (background is too white) because it makes it hard to interpret the data.

We thanks for the reviewer for the critical suggestion. We are very sorry for the mistake regarding supplement figure 1D, the VHL image was accidently lost when we saved the data for publication figure from the original western blots image. Please see the source data supplemental figure 1D. We also attached all of our original source western blot images to ensure our data consistency between the original western blots and organized figures. As suggested by the reviewers, we have adjusted the western blot images throughout the figures to make sure that they are consistent with their original format. Please check our updated figures.

2. In-vivo metastasis results are not clear, just ex-vivo BLI analysis is not sufficient to justify the metastatic role of ZHX2. It is well established that MDA-MB-231 cells highly metastasize to lungs (recent paper: PMID: 34465361). Authors were requested to provide H and E of whole lungs or a major lobe of lung to discern the metastatic effect. This way, authors can even highlight micro metastasis in inset of the figure.

We thank the comments from reviewer. We would like to perform H&E of the lung but unfortunately, we didn’t keep the lung tissues after the ex vivo imaging for further analysis so that we cannot perform this experiment at this time. We will save the lung tissues for the metastasis study in our future studies. We also want to note that in our previous and several other studies were using ex vivo BLI imaging to examine the metastasis, which showed robust and consistent results with other techniques (1-4). Again, we acknowledge the potential caveat that we did not further investigate the metastasis by other approaches in our current study.

Reviewer #3 (Recommendations for the authors):The authors have done a good job in addressing my prior concerns. I am satisfied with the responses to my criticisms/comments except one major question.Fig1F. ZHX2 expression pattern is opposite in several patients compared with previous version. Authors need to provide patients' information in supplemental tables.

We thank the comments from reviewer. In our initial western blot of the TNBC patient samples, we accidently loaded the marker together with one of the patient samples as shown in the Ponceau red staining. In his revision, we tried to rerun the WB samples however the Ponceau red staining suggested most of the proteins were degraded. Therefore, we freshly extracted new lysates from the same 10 pairs of patient frozen tissue samples and ran western blotting but in a different order of the sample organization from our initial version. Therefore, the expression pattern looks different from the initial version. We are sorry for making this confusion to the reviewers but we want to note that the overall expression of ZHX2 in these 10 pairs TNBC samples from biological replicate experiments is quite similar.

The clinical information of the included TNBC patients have been provided in the table Supplementary File 1b.

References:

1. Hu L, Xie H, Liu X, Potjewyd F, James LI, Wilkerson EM, et al. TBK1 Is a Synthetic Lethal Target in Cancer with VHL Loss. Cancer Discov 2020;10(3):460-75 doi 10.1158/2159-8290.CD-19-0837.

2. Zurlo G, Liu X, Takada M, Fan C, Simon JM, Ptacek TS, et al. Prolyl hydroxylase substrate adenylosuccinate lyase is an oncogenic driver in triple negative breast cancer. Nat Commun 2019;10(1):5177 doi 10.1038/s41467-019-13168-4.

3. Pein M, Insua-Rodriguez J, Hongu T, Riedel A, Meier J, Wiedmann L, et al. Metastasis-initiating cells induce and exploit a fibroblast niche to fuel malignant colonization of the lungs. Nat Commun 2020;11(1):1494 doi 10.1038/s41467-020-15188-x.

4. Taftaf R, Liu X, Singh S, Jia Y, Dashzeveg NK, Hoffmann AD, et al. ICAM1 initiates CTC cluster formation and trans-endothelial migration in lung metastasis of breast cancer. Nat Commun 2021;12(1):4867 doi 10.1038/s41467-021-25189-z.